# LightRAG: Simple and Fast Retrieval-Augmented Generation

## Abstract

Retrieval-Augmented Generation (RAG) systems enhance large language models (LLMs) by integrating external knowledge sources, enabling more accurate and contextually relevant responses tailored to user needs. However, existing RAG systems have significant limitations, including reliance on flat data representations and inadequate contextual awareness, which can lead to fragmented answers that fail to capture complex inter-dependencies. To address these challenges, we propose LightRAG, which incorporates graph structures into text indexing and retrieval processes. This innovative framework employs a dual-level retrieval system that enhances comprehensive information retrieval from both low-level and high-level knowledge discovery. Additionally, the integration of graph structures with vector representations facilitates efficient retrieval of related entities and their relationships, significantly improving response times while maintaining contextual relevance. This capability is further enhanced by an incremental update algorithm that ensures the timely integration of new data, allowing the system to remain effective in dynamic environments. Extensive experimental validation demonstrates considerable improvements in retrieval accuracy and efficiency compared to existing approaches. For the reproducibility of our results, we have made our LightRAG available anonymously at: https://anonymous.4open.science/r/LightRAG-2BEE.

## 1 Introduction

Retrieval-Augmented Generation (RAG) systems have been developed to enhance large language models (LLMs) by integrating external knowledge sources Sudhi et al. (2024); Es et al. (2024); Salemi & Zamani (2024). This innovative integration allows LLMs to generate more accurate and contextually relevant responses, significantly improving their utility in real-world applications. By adapting to specific domain knowledge Tu et al. (2024), RAG systems ensure that the information provided is not only pertinent but also tailored to the user's needs. Furthermore, they offer access to up-to-date information Zhao et al. (2024), which is crucial in rapidly evolving fields. Chunking plays a vital role in facilitating the retrieval-augmented generation process Lyu et al. (2024). By breaking down a large external text corpus into smaller, more manageable segments, chunking significantly enhances the accuracy of information retrieval. This approach allows for more targeted similarity searches, ensuring that the retrieved content is directly relevant to user queries.

However, existing RAG systems have key limitations that hinder their performance. **First**, many methods rely on flat data representations, restricting their ability to understand and retrieve information based on intricate relationships between entities. **Second**, these systems often lack the contextual awareness needed to maintain coherence across various entities and their interrelations, resulting in responses that may not fully address user queries. For example, consider a user asking, "How does the rise of electric vehicles influence urban air quality and public transportation infrastructure?" Existing RAG methods might retrieve separate documents on electric vehicles, air pollution, and public transportation challenges but struggle to synthesize this information into a cohesive response. They may fail to explain how the adoption of electric vehicles can improve air quality, which in turn could affect public transportation planning. As a result, the user may receive a fragmented answer that does not adequately capture the complex inter-dependencies among these topics.

To address these limitations, we propose incorporating graph structures into text indexing and relevant information retrieval. Graphs are particularly effective at representing the interdependencies

among different entities Rampášek et al. (2022), which enables a more nuanced understanding of relationships. The integration of graph-based knowledge structures facilitates the synthesis of information from multiple sources into coherent and contextually rich responses. Despite these advantages, developing a fast and scalable graph-empowered RAG system that efficiently handles varying query volumes is crucial. In this work, we achieve an effective and efficient RAG system by addressing three key challenges: i) **Comprehensive Information Retrieval**. Ensuring comprehensive information retrieval that captures the full context of inter-dependent entities from all documents; ii) **Enhanced Retrieval Efficiency**. Improving retrieval efficiency over the graph-based knowledge structures to significantly reduce response times; iii) **Rapid Adaptation to New Data**. Enabling quick adaptation to new data updates, ensuring the system remains relevant in dynamic environments.

In response to the outlined challenges, we propose LightRAG, a model that seamlessly integrates a graph-based text indexing paradigm with a dual-level retrieval framework. This innovative approach enhances the system's capacity to capture complex inter-dependencies among entities, resulting in more coherent and contextually rich responses. LightRAG employs efficient dual-level retrieval strategies: low-level retrieval, which focuses on precise information about specific entities and their relationships, and high-level retrieval, which encompasses broader topics and themes. By combining both detailed and conceptual retrieval, LightRAG effectively accommodates a diverse range of quries, ensuring that users receive relevant and comprehensive responses tailored to their specific needs. Additionally, by integrating graph structures with vector representations, our framework facilitates efficient retrieval of related entities and relations while enhancing the comprehensiveness of results through relevant structural information from the constructed knowledge graph.

In summary, the key contributions of this work are highlighted as follows:

- **General Aspect**. We emphasize the importance of developing a graph-empowered RAG system to overcome the limitations of existing methods. By integrating graph structures into text indexing, we can effectively represent complex interdependencies among entities, fostering a nuanced understanding of relationships and enabling coherent, contextually rich responses.

- **Methodologies**. To enable an efficient and adaptive RAG system, we propose LightRAG, which integrates a dual-level retrieval paradigm with graph-enhanced text indexing. This approach captures both low-level and high-level information for comprehensive, cost-effective retrieval. By eliminating the need to rebuild the entire index, LightRAG reduces computational costs and accelerates adaptation, while its incremental update algorithm ensures timely integration of new data, maintaining effectiveness in dynamic environments.

- **Experimental Findings**. Extensive experiments were conducted to evaluate the effectiveness of LightRAG in comparison to existing RAG models. These assessments focused on several key dimensions, including retrieval accuracy, model ablation, response efficiency, and adaptability to new information. The results demonstrated significant improvements over baseline methods.

## 2 RETRIEVAL-AUGMENTED GENERATION

Retrieval-Augmented Generation (RAG) integrates user queries with a collection of pertinent documents sourced from an external knowledge database, incorporating two essential elements: the **Retrieval Component** and the **Generation Component**. 1) The retrieval component is responsible for fetching relevant documents or information from the external knowledge database. It identifies and retrieves the most pertinent data based on the input query. 2) After the retrieval process, the generation component takes the retrieved information and generates coherent, contextually relevant responses. It leverages the capabilities of the language model to produce meaningful outputs. Formally, this RAG framework, denoted as $\mathcal{M}$, can be defined as follows:

$$\mathcal{M} = \Big( \mathcal{G}, \ \mathcal{R} = (\varphi, \psi) \Big), \quad \mathcal{M}(q; \mathcal{D}) = \mathcal{G}\Big( q, \psi(q; \hat{\mathcal{D}}) \Big), \quad \hat{\mathcal{D}} = \varphi(\mathcal{D}) \tag{1}$$

In this framework, $\mathcal{G}$ and $\mathcal{R}$ represent the generation module and the retrieval module, respectively, while $q$ denotes the input query and $D$ refers to the external database. The retrieval module $\mathcal{R}$ includes two key functionalities: i) **Data Indexer** $\varphi(\cdot)$: which involves building a specific data structure $\hat{\mathcal{D}}$ based on the external database $D$. ii) **Data Retriever** $\psi(\cdot)$: The relevant documents are obtained by comparing the query against the indexed data, also denoted as "relevant documents". By

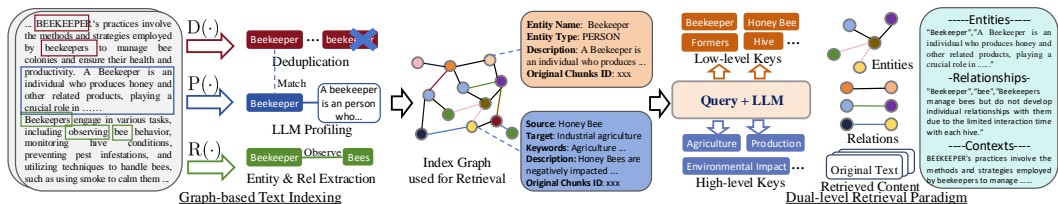

Figure 1: Overall architecture of the proposed LightRAG framework.

leveraging the information retrieved through $\psi(\cdot)$ along with the initial query $q$, the generative model $\mathcal{G}(\cdot)$ efficiently produces high-quality, contextually relevant responses.

In this work, we target several key points essential for an efficient and effective Retrieval-Augmented Generation (RAG) system which are elaborated below:

- **Comprehensive Information Retrieval**: The indexing function $\varphi(\cdot)$ must be adept at extracting global information, as this is crucial for enhancing the model's ability to answer queries effectively.

- **Efficient and Low-Cost Retrieval**: The indexed data structure $\hat{\mathcal{D}}$ must enable rapid and cost-efficient retrieval to effectively handle a high volume of queries.

- **Fast Adaptation to Data Changes**: The ability to swiftly and efficiently adjust the data structure to incorporate new information from the external knowledge base, is crucial for ensuring that the system remains current and relevant in an ever-changing information landscape.

## 3 THE LIGHTRAG ARCHITECTURE

### 3.1 GRAPH-BASED TEXT INDEXING

**Graph-Enhanced Entity and Relationship Extraction**. Our LightRAG enhances the retrieval system by segmenting documents into smaller, more manageable pieces. This strategy allows for quick identification and access to relevant information without analyzing entire documents. Next, we leverage LLMs to identify and extract various entities (e.g., names, dates, locations, and events) along with the relationships between them. The information collected through this process will be used to create a comprehensive knowledge graph that highlights the connections and insights across the entire collection of documents. We formally represent this graph generation module as follows:

$$\hat{\mathcal{D}} = (\hat{\mathcal{V}}, \hat{\mathcal{E}}) = \text{Dedupe} \circ \text{Prof}(\mathcal{V}, \mathcal{E}), \quad \mathcal{V}, \mathcal{E} = \cup_{\mathcal{D}_i \in \mathcal{D}} \text{Recog}(\mathcal{D}_i) \tag{2}$$

where $\hat{\mathcal{D}}$ represents the resulting knowledge graphs. To generate this data, we apply three main processing steps to the raw text documents $\mathcal{D}_i$. These steps utilize a LLM for text analysis and processing. Details about the prompt templates and specific settings for this part can be found in Appendix 7.3.2. The functions used in our graph-based text indexing paradigm are described as follows:

- **Extracting Entities and Relationships**. R($\cdot$): This function prompts a LLM to identify entities (nodes) and their relationships (edges) within the text data. For instance, it can extract entities like "Cardiologists" and "Heart Disease," and relationships such as "Cardiologists diagnose Heart Disease" from the text: "Cardiologists assess symptoms to identify potential heart issues." To improve efficiency, the raw text $\mathcal{D}$ is segmented into multiple chunks $\mathcal{D}_i$.

- **LLM Profiling for Key-Value Pair Generation**. P($\cdot$): We employ a LLM-empowered profiling function, P($\cdot$), to generate a text key-value pair $(K, V)$ for each entity node in $\mathcal{V}$ and relation edge in $\mathcal{E}$. Each index key is a word or short phrase that enables efficient retrieval, while the corresponding value is a text paragraph summarizing relevant snippets from external data to aid in text generation. Entities use their names as the sole index key, whereas relations may have multiple index keys derived from LLM enhancements that include global themes from connected entities.

- **Deduplication to Optimize Graph Operations**. D($\cdot$): Finally, we implement a deduplication function, D($\cdot$), that identifies and merges identical entities and relations from different segments of the raw text $\mathcal{D}_i$. This process effectively reduces the overhead associated with graph operations on $\hat{\mathcal{D}}$ by minimizing the graph's size, leading to more efficient data processing.

Our LightRAG offers two advantages through its graph-based text indexing paradigm. *First*, **Comprehensive Information Understanding**. The constructed graph structures enable the extraction of global information from multi-hop subgraphs, greatly enhancing LightRAG's ability to handle complex queries that span multiple document chunks. *Second*, **Enhanced Retrieval Performance**. the key-value data structures derived from the graph are optimized for rapid and precise retrieval. This provides a superior alternative to less accurate embedding matching methods (Gao et al., 2023) and inefficient chunk traversal techniques (Edge et al., 2024) commonly used in existing approaches.

**Fast Adaptation to Incremental Knowledge Base**. To efficiently adapt to evolving data changes while ensuring accurate and relevant responses, our LightRAG incrementally updates the knowledge base without the need for complete reprocessing of the entire external database. For a new document $\mathcal{D}'$, the incremental update algorithm processes it using the same graph-based indexing steps $\varphi$ as before, resulting in $\hat{\mathcal{D}}' = (\hat{\mathcal{V}}', \hat{\mathcal{E}}')$. Subsequently, LightRAGcombines the new graph data with the original by taking the union of the node sets $\hat{\mathcal{V}}$ and $\hat{\mathcal{V}}'$, as well as the edge sets $\hat{\mathcal{E}}$ and $\hat{\mathcal{E}}'$.

Two key objectives guide our approach to fast adaptation for the incremental knowledge base: **Seamless Integration of New Data**. By applying a consistent methodology to new information, the incremental update module allows the LightRAG to integrate new external databases without disrupting the existing graph structure. This approach preserves the integrity of established connections, ensuring that historical data remains accessible while enriching the graph without conflicts or redundancies. **Reducing Computational Overhead** . By eliminating the need to rebuild the entire index graph, this method reduces computational overhead and facilitates the rapid assimilation of new data. Consequently, LightRAG maintains system accuracy, provides current information, and conserves resources, ensuring users receive timely updates and enhancing the overall RAG effectiveness.

### 3.2 DUAL-LEVEL RETRIEVAL PARADIGM

To retrieve relevant information from both specific document chunks and their complex interdependencies, our LightRAG proposes generating query keys at both detailed and abstract levels.

- **Specific Queries**. These queries are detail-oriented and typically reference specific entities within the graph, requiring precise retrieval of information associated with particular nodes or edges. For example, a specific query might be, "Who wrote 'Pride and Prejudice'?"

- **Abstract Queries**. In contrast, abstract queries are more conceptual, encompassing broader topics, summaries, or overarching themes that are not directly tied to specific entities. An example of an abstract query is, "How does artificial intelligence influence modern education?"

To accommodate diverse query types, the LightRAG employs two distinct retrieval strategies within the dual-level retrieval paradigm. This ensures that both specific and abstract inquiries are addressed effectively, allowing the system to deliver relevant responses tailored to user needs.

- **Low-Level Retrieval**. This level is primarily focused on retrieving specific entities along with their associated attributes or relationships. Queries at this level are detail-oriented and aim to extract precise information about particular nodes or edges within the graph.

- **High-Level Retrieval**. This level addresses broader topics and overarching themes. Queries at this level aggregate information across multiple related entities and relationships, providing insights into higher-level concepts and summaries rather than specific details.

**Integrating Graph and Vectors for Efficient Retrieval**. By combining graph structures with vector representations, the model gains a deeper insight into the interrelationships among entities. This synergy enables the retrieval algorithm to effectively utilize both local and global keywords, streamlining the search process and improving the relevance of results.

- (i) **Query Keyword Extraction**. For a given query $q$, the retrieval algorithm of LightRAG begins by extracting both local query keywords $k^{(l)}$ and global query keywords $k^{(g)}$.

- (ii) **Keyword Matching**. The algorithm uses an efficient vector database to match local query keywords with candidate entities and global query keywords with relations linked to global keys.

- (iii) **Incorporating High-Order Relatedness**. To enhance the query with higher-order relatedness, LightRAGfurther gathers neighboring nodes within the local subgraphs of the retrieved graph

elements. This process involves the set $\{v_i | v_i \in \mathcal{V} \wedge (v_i \in \mathcal{N}_v \vee v_i \in \mathcal{N}_e)\}$, where $\mathcal{N}_v$ and $\mathcal{N}_e$ represent the one-hop neighboring nodes of the retrieved nodes $v$ and edges $e$, respectively.

This dual-level retrieval paradigm not only facilitates efficient retrieval of related entities and relations through keyword matching, but also enhances the comprehensiveness of results by integrating relevant structural information from the constructed knowledge graph.

### 3.3 Retrieval-Augmented Answer Generation

**Utilization of Retrieved Information**. Utilizing the retrieved information $\psi(q; \hat{\mathcal{D}})$, our LightRAG employs a general-purpose LLM to generate answers based on the collected data. This data comprises concatenated values $V$ from relevant entities and relations, produced by the profiling function $P(\cdot)$. It includes names, descriptions of entities and relations, and excerpts from the original text.

**Context Integration and Answer Generation**. By unifying the query with this multi-source text, the LLM generates informative answers tailored to the user's needs, ensuring alignment with the query's intent. This approach streamlines the answer generation process by integrating both context and query into the LLM model, as illustrated in detailed examples (Appendix 7.2).

### 3.4 Complexity Analysis of the LightRAG Framework

In this section, we analyze the complexity of our proposed LightRAG framework. The complexity is primarily divided into two parts. The first part is the graph-based Index phase. During this phase, we utilize the LLM to extract entities and relationships from each chunk of text. Thus, the LLM needs to be called $\frac{\text{total tokens}}{\text{chunk size}}$ times. Apart from this, there is no additional overhead, making this approach highly efficient in handling new text updates.

The second part is the graph-based retrieval phase. For each query, we first invoke the LLM to generate the relevant keywords. Similar to traditional RAG systems Gao et al. (2023; 2022); Chan et al. (2024), the retrieval process relies on vector-based search. However, instead of retrieving chunks as in traditional RAG, we focus on retrieving entities and relationships. Compared to the community-based traversal retrieval method used in GraphRAG, our approach significantly reduces the retrieval overhead.

## 4 Evaluation

We conduct empirical evaluations to validate the effectiveness of the proposed LightRAG framework. The experiments aim to evaluate LightRAG from the following dimensions: • How does LightRAG perform in comparison to existing RAG baseline methods? • How does the dual-level retrieval and the graph-based indexing benefit the generation quality of LightRAG? • What specific advantages does LightRAG exhibit in specific cases? • How is the cost and the adaptation abilities of LightRAG?

### 4.1 Experimental Settings

**Evaluation Datasets**. To conduct a comprehensive analysis of LightRAG, we selected four datasets from the UltraDomain benchmark (Qian et al., 2024). The UltraDomain data is sourced from 428 college textbooks, covering 18 distinct domains, including agriculture, social sciences, and humanities. From these domains, we chose the Agriculture, CS, Legal, and Mix datasets. The total number of tokens in each dataset ranges from 600,000 to 5,000,000, with detailed information provided in Table 4. Below is a specific introduction to the four domains used in our experiments:

- **Agriculture**: This domain focuses on agricultural practice, covering diverse topics including beekeeping, hive management, crop production, disease prevention, and so on.
- **CS**: This is the computer science domain. It encompasses key areas of data science and software engineering, particularly machine learning and big data processing, featuring content on recommendation systems, classification algorithms, and real-time analytics using Spark.
- **Legal**: It centers around corporate legal practices, including materials on corporate restructuring, legal agreements, regulatory compliance, and governance, targeting the legal and financial sectors.

- **Mixed**: This domain offers a diverse range of literary, biographical, and philosophical texts. It covers a wide spectrum of disciplines such as cultural, historical, and philosophical studies.

**Question Generation**. To evaluate the effectiveness of RAG systems for more high-level sensemaking tasks, we merge all the text content from each dataset as context and follow the generation method used in Edge et al. (2024). Specifically, we task an LLM with generating 5 RAG users and 5 tasks for each user. Each generated user is associated with a textual description for his/her expertise and traits that motivates his/her activities in raising questions. Each task of users is also described with texts, highlighting one of the user's potential intentions when interacting with RAG systems. For each user-task combination, the LLM is then required to generate 5 questions that necessitate an understanding of the entire corpus. In total, 125 questions are generated for each dataset.

**Baselines**. LightRAG is compared to the following four baseline methods across all datasets:

- **Naive RAG** (Gao et al., 2023): This model is a standard baseline in simple RAG systems. It segments raw texts into chunks and stores them in a vector database with text embeddings. For queries, Naive RAG creates vectorized representations to directly retrieve text chunks based on the highest similarity in their vector representations, ensuring efficient and straightforward matching.
- **RQ-RAG** (Chan et al., 2024): This approach utilizes the LLM to decompose the input query into several sub-queries. These sub-queries are designed to enhance search accuracy through explicit query rewriting, decomposition, and disambiguation.
- **HyDE** (Gao et al., 2022): This method employs the LLM to generate a hypothetical document based on the input query. The hypothetical document is then used to retrieve relevant text chunks, which are subsequently used to formulate the final answer.
- **GraphRAG** (Edge et al., 2024): This is a graph-enhanced RAG system which leverages an LLM to extract entities and relationships as nodes and edges from the text and generates corresponding descriptions for them. It aggregates nodes into communities and generates a community report to capture global information. When faced with high-level queries, GraphRAG retrieves more comprehensive information by traversing the communities.

**Implementation and Evaluation Details**. In our experiments, we use the *nano vector database* for vector data management and access. By default, GPT-4o-mini is used for all LLM-based operations in LightRAG. The chunk size across all datasets is set to 1200, and the gleaning parameter is fixed at 1 for both GraphRAG and our LightRAG.

As it is hard to define ground truth for many RAG queries, especially for complex queries involving high-level semantics, we follow existing work (Edge et al., 2024) and adopt the LLM-based multi-dimensional comparison method. A strong LLM, specifically GPT-4o-mini, is used to rank between each baseline and our LightRAG. The prompt we used for evaluation is elaborated in Appendix 7.3.4. Totally four evaluation dimensions are utilized, including:

i) **Comprehensiveness**: How much detail does the answer provide to cover all aspects and details of the question? ii) **Diversity**: How varied and rich is the answer in providing different perspectives and insights on the question? iii) **Empowerment**: How well does the answer help the reader understand and make informed judgments about the topic? iv) **Overall**: Considers the cumulative performance across the other three dimensions to determine the best overall answer.

The LLM directly compare two answers for each dimension and select the superior answer for each dimension. After determining the winning answer for each of the three dimensions, the LLM then combine the results across all dimensions to identify the overall better answer. To avoid any bias caused by the order of answers in the prompt during evaluation, we alternate the placement of each answer in the prompt and calculate win rates accordingly to obtain the final result.

## 4.2 Comparing LightRAG to Existing RAGs

We compare LightRAG against each baseline one-by-one on the four evaluation dimensions and four datasets. The results are reported in Table 1. From the results, we make the following conclusions:

**Advantage of Graph-based Retrieval in Large-Scale Token Queries**: When dealing with large token counts and complex queries requiring comprehensive consideration of the dataset's background,

Table 1: Win rates (%) of baselines v.s. LightRAG across four datasets and four evaluation dimensions.

| | Agriculture | | CS | | Legal | | Mix | |
|---|---|---|---|---|---|---|---|---|
| | NaiveRAG | **LightRAG** | NaiveRAG | **LightRAG** | NaiveRAG | **LightRAG** | NaiveRAG | **LightRAG** |
| Comprehensiveness | 32.69% | 67.31% | 35.44% | 64.56% | 19.05% | 80.95% | 36.36% | 63.64% |
| Diversity | 24.09% | 75.91% | 35.24% | 64.76% | 10.98% | 89.02% | 30.76% | 69.24% |
| Empowerment | 31.35% | 68.65% | 35.48% | 64.52% | 17.59% | 82.41% | 40.95% | 59.05% |
| Overall | 33.30% | 66.70% | 34.76% | 65.24% | 17.46% | 82.54% | 37.59% | 62.40% |
| | RQ-RAG | **LightRAG** | RQ-RAG | **LightRAG** | RQ-RAG | **LightRAG** | RQ-RAG | **LightRAG** |
| Comprehensiveness | 32.05% | 67.95% | 39.30% | 60.70% | 18.57% | 81.43% | 38.89% | 61.11% |
| Diversity | 29.44% | 70.56% | 38.71% | 61.29% | 15.14% | 84.86% | 28.50% | 71.50% |
| Empowerment | 32.51% | 67.49% | 37.52% | 62.48% | 17.80% | 82.20% | 43.96% | 56.04% |
| Overall | 33.29% | 66.71% | 39.03% | 60.97% | 17.80% | 82.20% | 39.61% | 60.39% |
| | HyDE | **LightRAG** | HyDE | **LightRAG** | HyDE | **LightRAG** | HyDE | **LightRAG** |
| Comprehensiveness | 24.39% | 75.61% | 36.49% | 63.51% | 27.68% | 72.32% | 42.17% | 57.83% |
| Diversity | 24.96% | 75.34% | 37.41% | 62.59% | 18.79% | 81.21% | 30.88% | 69.12% |
| Empowerment | 24.89% | 75.11% | 34.99% | 65.01% | 26.99% | 73.01% | 45.61% | 54.39% |
| Overall | 23.17% | 76.83% | 35.67% | 64.33% | 27.68% | 72.32% | 42.72% | 57.28% |
| | GraphRAG | **LightRAG** | GraphRAG | **LightRAG** | GraphRAG | **LightRAG** | GraphRAG | **LightRAG** |
| Comprehensiveness | 45.56% | 54.44% | 45.98% | 54.02% | 47.13% | 52.87% | 51.86% | 48.14% |
| Diversity | 19.65% | 80.35% | 39.64% | 60.36% | 25.55% | 74.45% | 35.87% | 64.13% |
| Empowerment | 36.69% | 63.31% | 45.09% | 54.91% | 42.81% | 57.19% | 52.94% | 47.06% |
| Overall | 43.62% | 56.38% | 45.98% | 54.02% | 45.70% | 54.30% | 51.86% | 48.14% |

Graph-based RAG systems like LightRAG and GraphRAG consistently outperform chunk-based retrieval methods (NaiveRAG, HyDE, and RQRAG). This performance gap becomes particularly evident as the dataset size increases. For instance, in the smallest dataset (Mix), the performance difference between chunk-based methods and LightRAG is relatively modest. However, in the largest dataset (Legal), the gap widens significantly, with chunk-based methods only achieving around 20% % win rates overall compared to LightRAG's dominance. This trend highlights the scalability advantage of Graph-based retrieval, especially in handling more extensive and intricate datasets.

**Consistent Superiority of LightRAG in the Diversity Dimension**: Compared to all kinds of baselines, LightRAG shows a noticeable edge in the Diversity metric, especially in the larger Legal dataset. Its consistent lead in Diversity suggests that it is more effective at generating a broader range of responses, particularly in scenarios where diverse content is critical. We attribute this advantage to the comprehensive information retrieval capabilities of LightRAG based on the effective graph-based text indexing, which always successfully fetches the full context for queries.

**LightRAG's Superiority over GraphRAG**: Although both LightRAG and GraphRAG leverage graph-based retrieval mechanisms, LightRAG consistently outperforms GraphRAG across multiple dimensions, particularly in larger and more complex datasets. In the Agriculture, CS, and Legal datasets, each containing millions of tokens, LightRAG achieved a clear and consistent lead, comprehensively surpassing GraphRAG. This demonstrates LightRAGś strong global control over the datasets, reinforcing its advantage in diverse and complex data environments. By generating keywords during both graph creation and query processing, LightRAG ensures that the graph remains extensible, efficiently handling a larger scope of data while maintaining performance.

## 4.3 ABLATION STUDIES

We further conduct ablation studies to validate the impact of our dual-level retrieval paradigm and the involvement of original text in our LightRAG. The results are presented in Table 2.

**Effectiveness of Dual-level Retrieval**. We first examine the impact of low-level and high-level retrieval. Two ablated models, each lacking one of the modules, are compared with LightRAG across four datasets. We make the following observations for different variants.

- **Low-level-only Retrieval**: The -High variant removes the high-order retrieval. This causes a significant performance drop for almost all datasets and metrics. This is mainly due to its low-level-only retrieval mechanism, which excessively focuses on entities and their direct neighbors. While allowing for deeper exploration of directly-related entities, it struggles to collect information for complex and difficult queries that require comprehensive insights.

Table 2: Performance of ablated versions of LightRAG, using NaiveRAG as reference.

| | Agriculture | | CS | | Legal | | Mix | |
|---|---|---|---|---|---|---|---|---|
| | NaiveRAG | **LightRAG** | NaiveRAG | **LightRAG** | NaiveRAG | **LightRAG** | NaiveRAG | **LightRAG** |
| Comprehensiveness | 32.69% | 67.31% | 35.44% | 64.56% | 19.05% | 80.95% | 36.36% | 63.64% |
| Diversity | 24.09% | 75.91% | 35.24% | 64.76% | 10.98% | 89.02% | 30.76% | 69.24% |
| Empowerment | 31.35% | 68.65% | 35.48% | 64.52% | 17.59% | 82.41% | 40.95% | 59.05% |
| Overall | 33.30% | 66.70% | 34.76% | 65.24% | 17.46% | 82.54% | 37.59% | 62.40% |
| | NaiveRAG | **-High** | NaiveRAG | **-High** | NaiveRAG | **-High** | NaiveRAG | **-High** |
| Comprehensiveness | 35.79% | 64.21% | 42.98% | 57.02% | 22.60% | 77.40% | 40.69% | 59.31% |
| Diversity | 26.86% | 73.14% | 35.09% | 64.91% | 16.09% | 83.91% | 37.15% | 62.84% |
| Empowerment | 35.02% | 64.98% | 42.98% | 57.02% | 23.58% | 76.42% | 48.26% | 51.74% |
| Overall | 35.33% | 64.67% | 42.98% | 57.02% | 21.92% | 78.08% | 41.39% | 58.61% |
| | NaiveRAG | **-Low** | NaiveRAG | **-Low** | NaiveRAG | **-Low** | NaiveRAG | **-Low** |
| Comprehensiveness | 36.31% | 63.69% | 41.82% | 58.18% | 18.92% | 81.08% | 35.92% | 64.08% |
| Diversity | 29.07% | 70.93% | 40.29% | 59.71% | 13.06% | 86.94% | 32.34% | 67.66% |
| Empowerment | 34.87% | 65.13% | 41.51% | 58.49% | 17.73% | 82.27% | 35.98% | 64.02% |
| Overall | 35.02% | 64.98% | 42.00% | 58.00% | 19.00% | 81.00% | 35.92% | 64.08% |
| | NaiveRAG | **-Origin** | NaiveRAG | **-Origin** | NaiveRAG | **-Origin** | NaiveRAG | **-Origin** |
| Comprehensiveness | 25.23% | 74.77% | 40.20% | 59.80% | 16.40% | 83.60% | 44.00% | 56.00% |
| Diversity | 26.94% | 73.06% | 45.36% | 54.64% | 13.15% | 86.84% | 26.00% | 74.00% |
| Empowerment | 31.01% | 68.99% | 42.61% | 57.39% | 18.55% | 81.45% | 45.33% | 54.67% |
| Overall | 25.51% | 74.49% | 40.20% | 59.80% | 17.12% | 82.88% | 44.00% | 56.00% |

- **High-level-only Retrieval**: The -Low variant focuses on capturing a broader range of content by leveraging entity-wise relationships instead of specific entities. This approach provides a clear advantage in the comprehensiveness dimension, as it is able to gather more extensive and varied information. However, the trade-off is a lack of depth in examining specific entities, which can limit its ability to provide highly detailed insights. As a result, this high-level-only retrieval method may struggle with tasks that demand more precise, detailed answers.

- **Hybird Mode**: The hybird mode, *i.e.* the full version of LightRAG, combines the strengths of both low-level and high-level modes, retrieving a broader set of relationships while simultaneously conducting an in-depth exploration of specific entities. This dual-level approach ensures both breadth in the retrieval process and depth in the analysis, offering a comprehensive view of the data. As a result, LightRAG achieves a balanced performance across multiple dimensions.

**Index Graph Excels in RAG**. We eliminated the use of original text in our retrieval process. Surprisingly, the resulting variant, -Origin, does not show significant performance declines across all four datasets. In some instances, this variant even shows performance improvements (e.g., in Agriculture and Mix). We attribute this phenomenon to the effective extraction of key information during the graph-based indexing process, which provides sufficient context for answering queries. Additionally, the original text is often highly coupled with less-relevant information, which can introduce noise when used to assist in answering questions.

## 4.4 CASE STUDY

To facilitate a clear comparison between baseline methods and our LightRAG, we present a specific case in Table 3, which includes responses to a machine learning question from both the competitive baseline, GraphRAG, and our LightRAG framework. In this instance, LightRAG outperforms in all evaluation dimensions assessed by the LLM judge, including comprehensiveness, diversity, empowerment, and overall quality. We make the following observations:

**i) Comprehensiveness**. Notably, LightRAG discusses a wider range of machine learning metrics, demonstrating its comprehensiveness and its ability to effectively interconnect all pertinent information. This showcases the strength of our graph-based indexing paradigm, which excels in precise entity and relation extraction and LLM profiling. **ii) Both Diversity and Empowerment**. Furthermore, LightRAG not only provides more diverse information but also more empowering content. This success can be attributed to LightRAG's dual-level retrieval paradigm, which conducts in-depth explorations focusing on related entities through low-level retrieval to enhance empowerment and broad explorations via high-level retrieval to improve answer diversity, capturing a comprehensive global perspective of the knowledge domain.

Table 3: A representative case comparing GraphRAG to our LightRAG method.

| Query | Which methods can normalize the feature values for effective machine learning? |
|---|---|
| **GraphRAG** | **Key Metrics for Evaluating Movie Recommendation Systems**: Evaluating the effectiveness of movie recommendation systems is crucial for ensuring user satisfaction and improving the overall performance of these systems. Several metrics are widely recognized for their ability to provide insights into how well a recommendation system performs. Below are the most informative metrics: 1. **Precision** [...] 2. **Recall** [...] 3. **F1 Score** [...] 4. **Mean Average Precision (MAP)** [...] 5. **Root Mean Squared Error (RMSE)** [...] 6. **User Satisfaction Metrics** [...] **Conclusion**: In summary, metrics such as precision, recall, F1 score, MAP, RMSE, and user satisfaction are critical for evaluating the performance of movie recommendation systems. Each metric provides unique insights that can help developers and researchers refine their algorithms and enhance user experiences. By focusing on these metrics, stakeholders can ensure that their recommendation systems effectively meet user preferences and expectations. |
| **LightRAG** | **Key Metrics for Evaluating Movie Recommendation Systems**: Evaluating movie recommendation systems involves multiple metrics that assess the effectiveness and accuracy of the recommendations provided to users. Below are some of the most informative metrics commonly used in this context, derived from the data and concepts surrounding recommendation systems. 1. **Mean Average Precision at K (MAPK)** [...] 2. **Precision and Recall** [...] 3. **Root Mean Squared Error (RMSE) and Mean Squared Error (MSE)** [...] 4. **Area Under the Curve (AUC)** [...] 5. **F-Measure** [...] 6. **User Engagement Metrics** [...] **Conclusion**: A comprehensive evaluation of movie recommendation systems employs a range of metrics that capture both the accuracy of the recommendations and user satisfaction. By effectively leveraging these metrics, system designers can optimize recommendation engines to deliver more personalized and relevant experiences to users. |
| **LLM Decision** | **Comprehensiveness**: "Winner": "Answer 2 (LightRAG)", "Explanation": "Answer 2 provides a broader array of metrics including MAPK, AUC, and user engagement metrics alongside precision, recall, RMSE, and F-Measure. This gives a fuller view of how recommendation systems can be evaluated. Answer 1, while detailed, is more focused on traditional metrics and lacks the inclusion of engagement metrics which are crucial for understanding user interaction." 

 **Diversity**: "Winner": "Answer 2 (LightRAG)", "Explanation": "Answer 2 not only covers a wide variety of metrics but also includes nuanced explanations of how some metrics interrelate and differ from one another, like the inclusion of both RMSE and MSE, as well as the details behind AUC. In contrast, Answer 1 sticks primarily to standard metrics without much exploration of potential nuances." 

 **Empowerment**: "Winner": "Answer 2 (LightRAG)", "Explanation": "Answer 2 empowers the reader more effectively by detailing how each metric functions and its importance in evaluating recommendation systems. By providing context such as the trade-offs between precision and recall and emphasizing user engagement metrics, it enables readers to make more informed judgments and understand the implications of different metrics. Answer 1 is more straightforward but lacks the depth of insight regarding why these metrics matter." 

 **Overall Winner**: "Winner": "Answer 2 (LightRAG)", "Explanation": "While Answer 1 is more direct and systematic, Answer 2 excels in comprehensiveness, diversity, and empowerment. It provides a richer exploration of the topic, including insights into user engagement and nuanced differences between metrics. This depth and breadth make it more informative for readers seeking to thoroughly understand the evaluation of movie recommendation systems." |

## 4.5 COST ANALYSIS

We compare the cost of our LightRAGwith that of the representative baseline GraphRAG across two key phases. Firstly, we compare the number of tokens and the number of API calls in the retrieval phase. Secondly, we compare the two metrics in the incremental text update process. The results of this evaluation on the legal dataset is shown in Table 2. Here $T_{\text{extract}}$ represents the token overhead for entity and relationship extraction, $C_{\text{max}}$ represents the maximum number of tokens allowed per API call, and $C_{\text{extract}}$ represents the number of API calls for entity and relationship extraction.

Figure 2: Cost in terms of number of tokens and API calls comparison for GraphRAG and LightRAG on the Legal dataset.

| Phase | Retrieval Phase | | Incremental Text Update | |
|---|---|---|---|---|
| Model | GraphRAG | Ours | GraphRAG | Ours |
| Tokens | $610 \times 1{,}000$ | $< 100$ | $1{,}399 \times 2 \times 5{,}000$ $+ T_{\text{extract}}$ | $T_{\text{extract}}$ |
| API Calls | $\frac{610 \times 1{,}000}{C_{\text{max}}}$ | 1 | $1{,}399 \times 2 + C_{\text{extract}}$ | $C_{\text{extract}}$ |

In the retrieval phase, GraphRAG generates 1,399 communities, of which 610 level-2 communities are actively used for retrieval in this experiment. Each community report contains an average of 1,000 tokens, resulting in a total token consumption of 610,000 tokens (610 communities × 1,000 tokens per community). Additionally, due to the need to traverse each community individually, hundreds of API calls are required, significantly increasing the retrieval overhead. In contrast, LightRAG optimizes this process by using less than 100 tokens for keyword generation and retrieval, which requires only a single API call for the entire retrieval process. This efficiency is achieved through our keyword-based retrieval mechanism, eliminating the need to process a large amount of information upfront.

In the incremental text update phase, while the entity and relationship extraction overhead is similar for both models, the handling of newly added data shows a stark difference. For a newly added dataset of the same size as the legal, GraphRAG must first deconstruct its existing community structure to incorporate new entities and relationships, and then regenerate the community structure. The token cost for generating each community report is approximately 5,000 tokens. Given the presence of 1,399 communities, GraphRAG would require approximately $1,399 \times 2 \times 5,000$ tokens to reconstruct both original and new community reports. In contrast, LightRAG only integrates newly extracted entities and relationships into the existing graph without needing to reconstruct the entire community structure, resulting in significantly lower overhead during incremental updates.

## 5 RELATED WORK

### 5.1 RETRIEVAL-AUGMENTED GENERATION WITH LLMS

Retrieval-Augmented Generation (RAG) systems enhance LLM inputs by retrieving relevant information from external sources, grounding responses in factual, domain-specific knowledge Ram et al. (2023); Fan et al. (2024). Current RAG approaches Gao et al. (2022; 2023); Chan et al. (2024); Yu et al. (2024) typically embed queries in a vector space to find the nearest context vectors. However, many of these methods rely on fragmented text chunks and only retrieve the top-k contexts, limiting their ability to capture comprehensive global information needed for effective responses.

Although recent studies Edge et al. (2024) have explored using graph structures for knowledge representation, two key limitations persist. First, these approaches often lack the capability for dynamic updates and expansions of the knowledge graph, making it difficult to incorporate new information effectively. In contrast, our proposed model, LightRAG, addresses these challenges by enabling the RAG system to quickly adapt to new information, ensuring the model's timeliness and accuracy. Additionally, existing methods often rely on brute-force searches for each generated community, which are inefficient for large-scale queries. Our LightRAG framework overcomes this limitation by facilitating rapid retrieval of relevant information from the graph through our proposed dual-level retrieval paradigm, significantly enhancing both retrieval efficiency and response speed.

### 5.2 LARGE LANGUAGE MODEL FOR GRAPHS

Graphs are a powerful framework for representing complex relationships and find applications in numerous fields. As Large Language Models (LLMs) continue to evolve, researchers have increasingly focused on enhancing their capability to interpret graph-structured data. This body of work can be divided into three primary categories: i) **GNNs as Prefix** where Graph Neural Networks (GNNs) are utilized as the initial processing layer for graph data, generating structure-aware tokens that LLMs can use during inference. Notable examples include GraphGPT Tang et al. (2024) and LLaGA Chen et al. (2024). ii) **LLMs as Prefix** involves LLMs processing graph data enriched with textual information to produce node embeddings or labels, ultimately refining the training process for GNNs, as demonstrated in systems like GALM Xie et al. (2023) and OFA Liu et al. (2024). iii) **LLMs-Graphs Integration** focuses on achieving a seamless interaction between LLMs and graph data, employing techniques such as fusion training and GNN alignment, and developing LLM-based agents capable of engaging with graph information directly Li et al. (2023); Brannon et al. (2023).

## 6 CONCLUSION

This work introduces an advancement in Retrieval-Augmented Generation (RAG) through the integration of a graph-based indexing approach that enhances both efficiency and comprehension in information retrieval. LightRAG utilizes a comprehensive knowledge graph to facilitate rapid and relevant document retrieval, enabling a deeper understanding of complex queries. Its dual-level retrieval paradigm allows for the extraction of both specific and abstract information, catering to diverse user needs. Furthermore, LightRAG's seamless incremental update capability ensures that the system remains current and responsive to new information, thereby maintaining its effectiveness over time. Overall, LightRAG excels in both efficiency and effectiveness, significantly improving the speed and quality of information retrieval and generation while reducing costs for LLM inference.

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

# 7 APPENDIX

## 7.1 EXPERIMENTAL DATA DETAILS

Table 4: Statistical information of the datasets.

| Statistics | Agriculture | CS | Legal | Mix |
|---|---|---|---|---|
| Total Documents | 12 | 10 | 94 | 61 |
| Total Tokens | 2,017,886 | 2,306,535 | 5,081,069 | 619,009 |

Table 4 presents statistical information for four datasets: Agriculture, CS, Legal, and Mix. The Agriculture dataset consists of 12 documents totaling 2,017,886 tokens, while the CS dataset contains 10 documents with 2,306,535 tokens. The Legal dataset is the largest, comprising 94 documents and 5,081,069 tokens. Lastly, the Mix dataset includes 61 documents with a total of 619,009 tokens.

## 7.2 RETRIEVAL AND GENERATION EXAMPLE

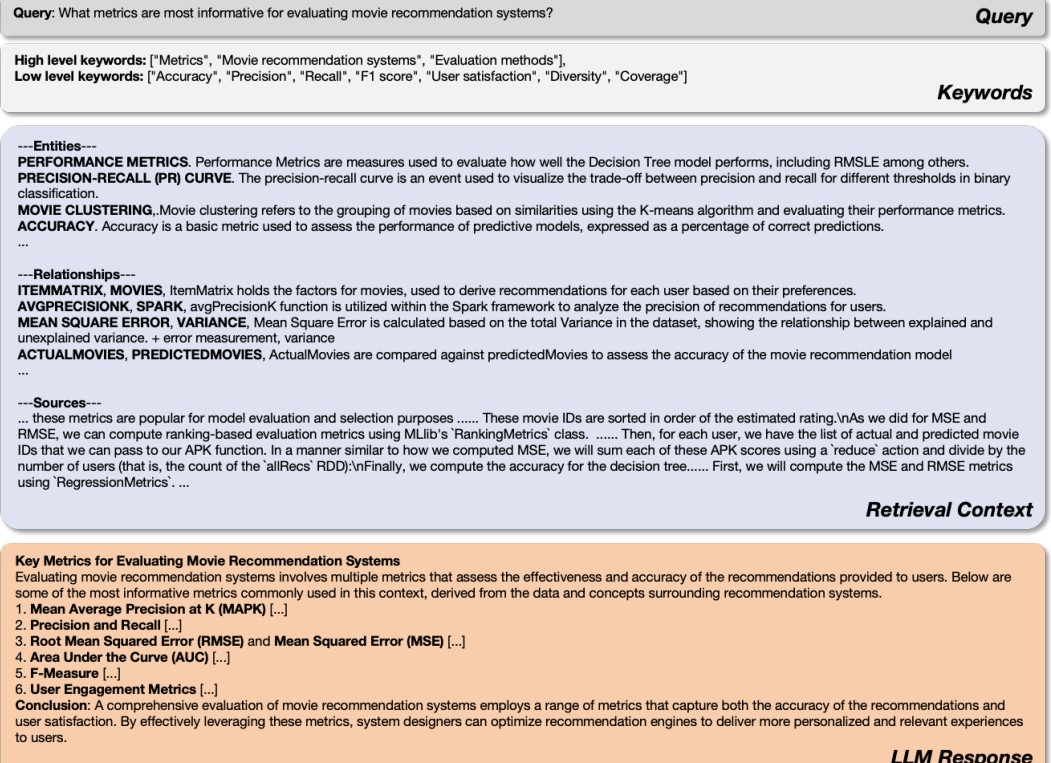

Figure 3: A retrieval and generation example.

In Figure 3, we illustrate a retrieve-and-generate process. Given the query "What metrics are most informative for evaluating movie recommendation systems?", the LLM first extracts both low-level and high-level keywords. These keywords are then used to guide the retrieval process on the graph, focusing on relevant entities and relationships. The retrieved information is organized into three components: entities, relationships, and sources, which are subsequently fed back into the LLM to generate a comprehensive answer to the query.

```
-Goal-
Given a text document that is potentially relevant to this activity and a list of entity types, identify all entities of those types from the text and all relationships among the
identified entities.

-Steps-
1. Identify all entities. For each identified entity, extract the following information:
- entity_name: Name of the entity, capitalized
- entity_type: One of the following types: [organization, person, geo, event]
- entity_description: Comprehensive description of the entity's attributes and activities
Format each entity as ("entity"<|><entity_name><|><entity_type><|><entity_description>)

2. From the entities identified in step 1, identify all pairs of (source_entity, target_entity) that are *clearly related* to each other.
For each pair of related entities, extract the following information:
- source_entity: name of the source entity, as identified in step 1
- target_entity: name of the target entity, as identified in step 1
- relationship_description: explanation as to why you think the source entity and the target entity are related to each other
- relationship_strength: a numeric score indicating strength of the relationship between the source entity and target entity
- relationship_keywords: one or more high-level key words that summarize the overarching nature of the relationship, focusing on concepts or themes rather than
specific details
Format each relationship as ("relationship"<|><source_entity><|><target_entity><|><relationship_description><|><relationship_keywords><|><relationship_strength>)

3. Identify high-level key words that summarize the main concepts, themes, or topics of the entire text. These should capture the overarching ideas present
in the document.
Format the content-level key words as ("content_keywords"<|><high_level_keywords>)

4. Return output in English as a single list of all the entities and relationships identified in steps 1 and 2. Use **##** as the list delimiter.

5. When finished, output <|COMPLETE|>

-Real Data-
Entity_types: {entity_types}
Text: {input_text}

Output:                                                                                                Graph Construct Prompt
```

Figure 4: Query Generate Prompt

## 7.3 PROMPTS

### 7.3.1 GRAPH CONSTRUCT PROMPT

The graph construction prompt outlined in Figure 4 is designed to extract and structure entity-relationship information from a text document based on specified entity types. The process begins by identifying entities and categorizing them into types such as organization, person, location, and event. It then provides detailed descriptions of their attributes and activities. Next, the prompt identifies relationships between these entities, offering explanations, assigning strength scores, and summarizing the relationships using high-level keywords.

### 7.3.2 QUERY GENERATE PROMPT

```
Given the following description of a dataset:  {total_description}

Please identify 5 potential users who would engage with this dataset. For each user, list 5 tasks they would perform with this dataset. Then, for each (user, task)
combination, generate 5 questions that require a high-level understanding of the entire dataset.

Output the results in the following structure:
   - User 1: [user description]
     - Task 1: [task description] [ Question 1: {Question 1}, Question 2: {Question 2}, Question 3: {Question 3}, Question 4: {Question 4}, Question 5: {Question 5} ]
     - Task 2: [task description] [ Question 1: {Question 1}, Question 2: {Question 2}, Question 3: {Question 3}, Question 4: {Question 4}, Question 5: {Question 5} ]
     ...
     - Task 5: [task description] [ Question 1: {Question 1}, Question 2: {Question 2}, Question 3: {Question 3}, Question 4: {Question 4}, Question 5: {Question 5} ]
   - User 2: [user description]
     ...
   - User 5: [user description]
     ...                                                                                                  Query Generate Prompt
```

Figure 5: Query Generate Prompt

In Figure 5, the query generation prompt outlines a framework for identifying potential user roles (e.g., data scientist, finance analyst, and product manager) and their objectives for generating queries based on a specified dataset description. The prompt explains how to define five distinct users who would benefit from interacting with the dataset. For each user, it specifies five key tasks they would perform while working with the dataset. Additionally, for each (user, task) combination, five high-level questions are posed to ensure a thorough understanding of the dataset.

```
---Role---
You are a helpful assistant tasked with identifying both high-level and low-level keywords in the user's query.

---Goal---
Given the query, list both high-level and low-level keywords. High-level keywords focus on overarching concepts or themes, while low-level keywords focus on
specific entities, details, or concrete terms.
```
**Keywords Generate Instruction Prompt**

```
- Output the keywords in JSON format.
- The JSON should have two keys:
- "high_level_keywords" for overarching concepts or themes.
- "low_level_keywords" for specific entities or details.

-Examples-

Example 1:
Query: "How does international trade influence global economic stability?"
Output: {{ "high_level_keywords": ["International trade", "Global economic stability", "Economic impact"], "low_level_keywords": ["Trade agreements", "Tariffs",
"Currency exchange", "Imports", "Exports"] }}

Example 2:
Query: "What are the environmental consequences of deforestation on biodiversity?"
Output: {{ "high_level_keywords": ["Environmental consequences", "Deforestation", "Biodiversity loss"], "low_level_keywords": ["Species extinction", "Habitat
destruction", "Carbon emissions", "Rainforest", "Ecosystem"] }}

Example 3:
Query: "What is the role of education in reducing poverty?"
Output: {{ "high_level_keywords": ["Education", "Poverty reduction", "Socioeconomic development"], "low_level_keywords": ["School access", "Literacy rates", "Job
training", "Income inequality"]  }}

-Real Data-
Query: {query}
Output:
```
**Keywords Generate Input Prompt**

Figure 6: Keywords Generate Prompt

### 7.3.3 KEYWORDS GENERATE PROMPT

In Figure 6, the prompt describes a method for extracting keywords from a user's query, distinguishing between high-level and low-level keywords. High-level keywords represent broad concepts or themes, while low-level keywords focus on specific entities and details. The extracted keywords are returned in JSON format, organized into two fields: "high_level_keywords" for overarching ideas and "low_level_keywords" for specific details.

### 7.3.4 EVALUATION PROMPT

```
---Role---
You are an expert tasked with evaluating two answers to the same question based on four criteria: Comprehensiveness, Diversity, and Empowerment.

---Goal---
You will evaluate two answers to the same question based on four criteria: Comprehensiveness, Diversity, and Empowerment.

- Comprehensiveness: How much detail does the answer provide to cover all aspects and details of the question?
- Diversity: How varied and rich is the answer in providing different perspectives and insights on the question?
- Empowerment: How well does the answer help the reader understand and make informed judgments about the topic?

For each criterion, choose the better answer (either Answer 1 or Answer 2) and explain why. Then, select an overall winner based on these three categories.
```
**Evaluation Instruction Prompt**

```
Here is the question: {query}

Here are the two answers: Answer 1: {answer1}; Answer 2:  {answer2}

Evaluate both answers using the three criteria listed above and provide detailed explanations for each criterion.

Output your evaluation in the following JSON format:

{{
    "Comprehensiveness": {{ "Winner": "[Answer 1 or Answer 2]", "Explanation": "[Provide explanation here]" }},
    "Diversity": {{ "Winner": "[Answer 1 or Answer 2]", "Explanation": "[Provide explanation here]" }},
    "Empowerment": {{ "Winner": "[Answer 1 or Answer 2]", "Explanation": "[Provide explanation here]" }},
    "Overall Winner": {{ "Winner": "[Answer 1 or Answer 2]", "Explanation": "[Summarize why this answer is the overall winner based on the three criteria]" }}
}}
```
**Evaluation Input Prompt**

Figure 7: Evaluation Prompt

The evaluation prompt is illustrated in Figure 7. It introduces a comprehensive evaluation framework for comparing two answers to the same question based on three key criteria: Comprehensiveness, Diversity, and Empowerment. Its purpose is to guide the LLM through the process of selecting the better answer for each criterion, followed by an overall assessment. For each of the three criteria, the

LLM must identify which answer performs better and provide a rationale for its choice. Ultimately, an overall winner is determined based on performance across all three dimensions, accompanied by a detailed summary that justifies the decision. The evaluation is structured in JSON format, ensuring clarity and consistency, and facilitating a systematic comparison between the two answers.

