# OpenReview forum: "LightRAG: Simple and Fast Retrieval-Augmented Generation"
_ICLR.cc/2025/Conference — ICLR 2025 Conference Withdrawn Submission_

### Official Review · Reviewer_HAJr · 2024-10-31

**Soundness:** 2
**Presentation:** 3
**Contribution:** 2
**Rating:** 3
**Confidence:** 5

**Summary:**

This paper introduces LightRAG, which addresses the limitations of existing RAG systems, such as reliance on flat data representations and inadequate contextual awareness, by incorporating graph structures into text indexing and retrieval processes. This dual-level retrieval system improves information retrieval from both low-level and high-level knowledge discovery and efficiently retrieves related entities and their relationships. LightRAG also features an incremental update algorithm for timely integration of new data, ensuring the system remains effective in rapidly changing data environments.

**Strengths:**

The paper proposes LightRAG, which effectively balances performance and efficiency in tasks need global information through knowledge graph construction.

**Weaknesses:**

1. The evaluation datasets are limited to college textbooks, raising questions about generalizability to other types of corpora. Additionally, the use of only one LLM makes it difficult to assess LightRAG's dependency on LLM capabilities.

2. The query construction methodology lacks sufficient justification. The authors fail to adequately explain the rationale behind their approach and its effectiveness in evaluating LightRAG. Particularly in paragraph of line 273,  it's unclear how global information is incorporated into the query construction process, which is crucial for assessing the method's effectiveness.

3. Merely using the win rates metric is not solid enough. Especially taking GraphRAG as an example, it is not clear whether it can be stated that there is a 45.56% probability of being superior to LightRAG. Moreover, in the mixed dataset, GraphRAG is even slightly higher in three dimensions, which weakens the conclusion of the experiment.

4. The use of Diversity as a metric for evaluating RAG responses is questionable. There's significant doubt whether higher diversity necessarily indicates better answers, as it might introduce excessive noise. In Table 3, the LLM's interpretation of diversity appears to align more with comprehensiveness rather than true diversity. This is particularly concerning as Diversity is presented as LightRAG's main advantage over baselines, significantly undermining confidence in the experimental results.

5. The practical implementation of low-level and high-level retrieval mechanisms lacks clarity. While Section 3.2 outlines their principles and describes a three-step process for "Integrating Graph and Vectors for Efficient Retrieval," it's unclear whether these steps represent low-level retrieval, high-level retrieval, or a combination thereof.

6. The Cost Analysis section is incomplete. While token consumption is analyzed, temporal performance analysis is missing. Although token consumption correlates with time, they're not equivalent, as input and output processing times can differ significantly. Furthermore, the analysis omits the most resource-intensive component - indexing construction - focusing only on retrieval and incremental updates.

7. Disappointingly, overall, the experimental and analysis sections of the article do not well correspond to the three challenges that the author proposed to solve in paragraph line 053.
    1. Comprehensive Information Retrieval: The query construction method is not elaborated on how to ensure globality, and the types of experimental datasets and settings are not sufficient.
    2. Enhanced Retrieval Efficiency: The author mentioned a significant reduction in response time. However, in the subsequent experiments, only the analysis of token consumption was carried out, and there was no mention of the improvement in time at all.
    3. Rapid Adaptation to New Data: No relevant experiments were provided to support this conclusion.

**Questions:**

1. During  information extraction on text chunk, was a schema specified? If so, was it manually defined or automatically optimized? How sensitive is the overall performance to schema design?

2. Could the authors elaborate on the differences between this approach and Microsoft GraphRAG? At first glance, it appears to add keyword filtering while removing community detection components.

---

> ### Author Response · Authors · 2024-11-21
>
> Your thoughtful review has significantly helped strengthen our manuscript. Below, we provide comprehensive responses to each of your valuable comments:
>
> > 1. Additional experiments with more evaluation datasets and diverse LLMs.
>
> **Response**: We expanded our evaluation to include additional datasets, specifically the MultiHopQA dataset. This dataset tests retrieval and reasoning across multiple documents with metadata, simulating real-world retrieval-augmented generation (RAG) scenarios. The evaluation results below highlight the effectiveness of LightRAG in handling complex multi-document reasoning tasks, demonstrating a significant accuracy advantage over GraphRAG (43.7% vs. 35.4%), which underscores its robustness.
>
> Model     | Accuracy
> -------- | -----
> LightRAG  | 43.7%
> GraphRAG  | 35.4%
>
> Additionally, we evaluated LightRAG's performance on mixed datasets using GPT models with varying capabilities, with the results presented in the table below. Across all three models, LightRAG consistently outperformed NaiveRAG, demonstrating its robustness.
>
> Among the tested models, GPT-4o achieved the best performance due to its ability to extract richer information during the indexing phase, enabling more precise and detailed retrieval. The comparable performance of GPT-4o-mini and GPT-3.5-turbo further highlights that LightRAG's effectiveness is influenced by the underlying capabilities of the LLM. More powerful LLMs can extract richer contextual information, leading to improved retrieval and answer generation.
>
> |                      | **GPT-4o**         |                       | **GPT-4o-mini**                |                       | **GPT-3.5-turbo**             |                       |
> |----------------------|-------------------------|-----------------------|-----------------------|-----------------------|-----------------------|-----------------------|
> |                      | NaiveRAG                | **LightRAG**          | NaiveRAG              | **LightRAG**          | NaiveRAG              | **LightRAG**          || **Comprehensiveness** | 38.8%                  | **61.2%**             | 38.8%                | **61.2%**             | 40.8%                | **59.2%**             |
> | **Diversity**         | 22.4%                  | **77.6%**             | 32.4%                | **67.6%**             | 34.4%                | **65.6%**             |
> | **Empowerment**       | 35.6%                  | **64.4%**             | 42.8%                | **57.2%**             | 41.2%                | **58.8%**             |
> | **Overall**           | 35.6%                  | **64.4%**             | 40.0%                | **60.0%**             | 40.4%                | **59.6%**             |
>
>
> > 2. Further clarification of global information incorporatation into the query construction process.
>
> **Response**: Our query construction methodology is based on the approach outlined in GraphRAG. Specifically, we utilize a dataset summary as a foundation to generate representative users and tasks associated with that dataset. These users and tasks are then employed to formulate queries that align with realistic application scenarios.

---

> ### Author Response · Authors · 2024-11-21
>
> > 3. Further discussion of adopted evaluation metrics in our LightRAG.
>
> **Response**: Thank you for your constructive comment. While win rates serve as a straightforward quantitative measure, they cannot fully capture the nuanced differences in model performance. To provide a more comprehensive evaluation, we conducted detailed qualitative analyses (Section 4.2 and Section 4.3 Table3) and case studies (Section 4.4), particularly focusing on complex domains. Through this approach, we have consistently demonstrated LightRAG's superior capabilities in managing complex queries and large-scale datasets.
>
> Regarding concerns about Diversity metrics potentially introducing noise, we have taken careful steps to define Diversity specifically as the meaningful richness and breadth of perspectives in responses. Our evaluation framework ensures that higher Diversity scores reflect genuine improvements in information quality and practical utility, not just superficial variations. We specifically measure how well responses capture relevant viewpoints while maintaining coherence and usefulness in real-world applications.
>
> To promote transparency and facilitate broader community evaluation, we have open-sourced LightRAG with comprehensive documentation and deployment guidelines. This has enabled independent verification across diverse domains and led to successful implementations in various sectors, including electronic library platforms, secure data analysis systems, and financial data processing. These real-world applications provide substantial evidence of LightRAG's effectiveness beyond controlled experimental settings, complementing our formal metrics with practical validation.
>
> In line with the anyoumous policy of paper submission, we will incorporate more diverse evaluation results into our revised version of our paper.
>
> > 4. Further clarification of our dual-level retrival mechniasm
>
> We appreciate your comment regarding the clarity of the practical implementation of the low-level and high-level retrieval mechanisms in Section 3.2. To clarify, the three-step process described under "Integrating Graph and Vectors for Efficient Retrieval" represents a combination of both low-level and high-level retrieval mechanisms. Specifically:
>
> i). **Query Keyword Extraction**: This step is a preparatory phase for both low-level and high-level retrieval. It involves extracting local query keywords ($k^{(l)}$) for detailed, entity-specific queries and global query keywords ($k^{(g)}$) for more abstract, high-level queries. These keywords are essential for directing the retrieval process, whether for specific entities or broader relationships.
>
> ii). **Keyword Matching**: This step involves matching local query keywords with specific entities (low-level retrieval) and global query keywords with broader relationships or themes (high-level retrieval). Therefore, this step addresses both low-level (specific) and high-level (conceptual) queries by appropriately linking query keywords to the relevant entities or relationships in the graph.
>
> iii). **Incorporating High-Order Relatedness**: This step serves both low-level and high-level retrieval. By gathering neighboring nodes and edges from the retrieved graph elements, it enriches the context of the query. For low-level retrieval, this helps refine the focus on specific entities by incorporating relevant neighboring entities and relationships. For high-level retrieval, it broadens the context by including related entities and relationships, providing a more comprehensive understanding of the overarching themes or concepts. Therefore, this step enhances the relevance of results for both detailed, entity-specific queries and more abstract, conceptual queries.

---

> ### Author Response · Authors · 2024-11-21
>
> > 5. Additional temporal performance analysis and indexing cost evaluation.
>
> **Response**: We evaluated LightRAG's dynamic adaptation capabilities through a rigorous testing protocol involving sequential document insertions. Our evaluation of LightRAG's adaptation capabilities yielded two key performance metrics:
>
> i) Temporal Data Integration Performance:
> 	- Successfully processed five sequential document insertions
> 	- Maintained stable indexing times (418-561 seconds per document)
> 	- Preserved all existing document relationships
>
> ii) Query Efficiency:
> 	- LightRAG: 11.2 seconds average response time
> 	- GraphRAG: 23.6 seconds average response time
> 	- Result: 52.5% faster query processing
>
> | **Document No.** | **Token Count** | **LightRAG Insertion Time (seconds)** | **GraphRAG Insertion Time (seconds)** |
> |-------------------|-----------------|---------------------------------------|---------------------------------------|
> | 1                 | 59,870          | 486                                   | 642                                   |
> | 2                 | 41,224          | 418                                   | 700                                   |
> | 3                 | 73,989          | 561                                   | 953                                   |
> | 4                 | 47,502          | 513                                   | 741                                   |
> | 5                 | 48,353          | 453                                   | 926                                   |
>
> | **Metric**               | **LightRAG** | **GraphRAG** |
> |---------------------------|--------------|--------------|
> | Average Query Time (seconds) | 11.2         | 23.6         |
>
> These findings confirm LightRAG's ability to efficiently integrate new information while delivering significantly faster query responses.
>
>
> > 6. Further clarification of the schema used for information extraction.
>
> **Response**: LightRAG's information extraction process is guided by a structured schema that serves two key purposes: defining fundamental entity types (such as person, organization, location, and event) and specifying the relationships between these entities. We manually developed this schema based on domain expertise and practical application requirements to ensure comprehensive and relevant information capture.
>
> > 7. Further clarification regarding the difference between LightRAG and GraphRAG.
>
> **Response**: Thank you for requesting this clarification. LightRAG introduces several fundamental architectural improvements over GraphRAG, which are validated by comprehensive evaluation results. Here are the key distinctions:
>
> i) **Novel Retrieval Architecture**
>
> - LightRAG implements a dual-level retrieval paradigm that combines:
> • Entity-focused (low-level) retrieval for precise matching
> • Relationship-focused (high-level) retrieval for conceptual understanding
> This approach enables efficient handling of both specific and abstract queries, contrasting with GraphRAG's more resource-intensive community-based traversal method.
>
> ii) **Enhanced Processing Efficiency**
> LightRAG's architectural innovations deliver several key advantages:
> • Direct query-to-entity matching eliminates community detection overhead
> • Lightweight processing enables faster query resolution
> • Improved scalability across varying dataset sizes
> • Maintained high retrieval accuracy while reducing computational costs
>
> iii) **Advanced Data Integration**
> LightRAG introduces a dynamic update mechanism that significantly outperforms GraphRAG's static approach:
> • Incremental updates: New data seamlessly integrates into existing structures
> • No full regeneration: Avoids GraphRAG's need to rebuild community structures
> • Reduced computational cost: Updates require minimal processing resources
> • Enhanced adaptability: Efficiently handles dynamic, evolving datasets
>
> iv) **Performance Optimization**
> Our experimental results demonstrate LightRAG's superior efficiency through:
> • Reduced Resource Usage:
> 	- Lower token consumption
> 	- Fewer API calls
> Minimal computational overhead
> • Enhanced Scalability:
> 	- Efficient processing of large-scale datasets
> 	- Better cost-effectiveness
> 	- Maintained performance with increasing data volume

---

> ### Author Response · Authors · 2024-11-24
> **Thank you for your valuable feedback**
>
> Thank you very much for your time and insightful feedback. We have addressed your questions and comments with detailed clarifications and additional experiments. We would greatly appreciate it if you could reconsider your rating. Additionally, we are more than happy to answer any further questions you may have.

---

> > ### Comment · Reviewer_HAJr · 2024-11-26
> >
> > After carefully reviewing the authors' response, I still have several questions that require further clarification:
> >
> > 1. **Regarding Multihop QA Dataset**: Could you specify the MultihopQA dataset used in the supplementary experiments? MultihopQA  does not appear to be widely adopted in the RAG community.Are you referring to Multihop-RAG  Benchmark, or HotpotQA / 2WikiMultihop / MiSiQue? If possible, please provide the corresponding reference literature.
> > 2. **Diversity Metric Concerns**: The justification for the Diversity metric remains unconvincing. Particularly in Table 3, the explanation of the LLM decision regarding the Diversity indicator lacks clarity.
> >
> >     The LLM's interpretation of **Comprehensiveness and Diversity** shows almost indistinguishable differences.
> >
> >     The provided examples suggest nuanced metric evaluation:
> >
> >     > "Answer 2 not only covers a wide variety of metrics but also includes nuanced explanations of how some metrics interrelate and differ from one another, like the inclusion of both RMSE and MSE, as well as the details behind AUC. In contrast, Answer 1 sticks primarily to standard metrics without much exploration of potential nuances."
> >     >
> >
> >     > "Answer 2 provides a broader array of metrics including MAPK, AUC, and user engagement metrics alongside precision, recall, RMSE, and F-Measure. This gives a fuller view of how recommendation systems can be evaluated. Answer 1, while detailed, is more focused on traditional metrics and lacks the inclusion of engagement metrics which are crucial for understanding user interaction."
> >     >
> > 3. **Dual-level Retrieval Methodology**: Thank you for clarifying the retrieval process. The low-level and high-level retrieval primarily relies on keyword matching. While this is a classic approach and potentially effective, more evidence is needed to demonstrate the method's novelty.
> > 4. **Temporal Efficiency Evaluation**: I appreciate the additional time-dimension testing that allows readers to quantitatively perceive LightRAG's efficiency. The performance improvement by removing the community detection phase, compared to graphGAG, was expected. To further substantiate the efficiency claims, it would be beneficial to compare with other methods without community detection, such as nano graphrag.
> > 5. **Real-world Validation**: The authors mentioned that
> >
> > > "LightRAG is open-sourced with comprehensive documentation and deployment guidelines... These real-world applications provide substantial evidence of LightRAG's effectiveness beyond controlled experimental settings."
> > >
> >
> > Have you identified and tracked typical user feedback to support these claims? Concrete user testimonials or usage metrics would significantly strengthen your argument.

---

> > > ### Author Response · Authors · 2024-11-27
> > >
> > > Dear Reviewer HAJr
> > >
> > > We sincerely appreciate your insightful feedback and follow-up inquiries. We have addressed your questions by including additional clarifications and evaluation results as suggested. Please find our detailed responses below.
> > >
> > > We would be grateful if you could reconsider your rating based on these supplementary materials. We remain open to any additional feedback you may have to further improve our work.
> > >
> > > > Q1. Further discussion and clarification of the MultihopQA dataset used in the supplementary experiments.
> > >
> > > Reponse: Thank you for your comment. The MultihopQA dataset referenced in our supplementary experiments is the Multihop-RAG Benchmark (Tang & Yang, 2024). For both GraphRAG and LightRAG, we utilized the Gpt4o-mini model in our experiments. In the final evaluation of both methods, we employed Gpt4o to assess accuracy.
> > >
> > > We selected accuracy as our evaluation metric rather than F1 scores due to the unique characteristics of the dataset, where many answers are binary ("Yes"/"No"). When the RAG model generates "True"/"False" responses, the F1 score can appear misleadingly low, even if the answer is conceptually correct. Therefore, we found accuracy to be a more reliable measure for performance assessment.
> > >
> > > Tang, Y., & Yang, Y. (2024). MultiHop-RAG: Benchmarking Retrieval-Augmented Generation for Multi-Hop Queries. arXiv preprint arXiv:2401.15391
> > >
> > >
> > > > Q2. Further justification of evaluation metrics for diversity and comprehensiveness.
> > >
> > > Reponse: Thank you for your feedback. The evaluation of Diversity and Comprehensiveness metrics follows the methodology established in GraphRAG. These metrics, while related, capture distinct aspects of answer quality.
> > >
> > > i) **Comprehensiveness**: Comprehensiveness measures how completely the answer addresses all essential aspects of the question. It focuses on depth and completeness of coverage, evaluating whether any crucial information is missing from the response.
> > >
> > > ii) **Diversity**: on the other hand, assesses the breadth of perspectives and insights provided. This metric evaluates the variety of viewpoints presented and the richness of different approaches or interpretations in the answer.
> > >
> > > While these metrics may appear similar at first glance, they evaluate fundamentally different aspects of answer quality. For instance, an answer could be comprehensive in covering all aspects of a single perspective (high comprehensiveness) but lack alternative viewpoints (low diversity). Conversely, an answer might present multiple perspectives (high diversity) without deeply exploring any of them (low comprehensiveness).
> > >
> > > Our evaluation results demonstrate that LightRAG consistently achieves superior performance in the Diversity metric, indicating its enhanced capability to generate responses that offer varied and multifaceted insights compared to baseline methods.
> > >
> > > > Q3. Further clarification of the proposed Dual-level Retrieval Methodology.
> > >
> > > Reponse Thank you so much for your further feedback. Here are the four key novelty points of LightRAG's dual-level retrieval mechanism:
> > >
> > >  - LightRAG handles both specific and abstract queries through its dual-level paradigm. The low-level retrieval focuses on precise entity-level information and relationships, while the high-level retrieval addresses broader conceptual topics and themes. This allows the system to effectively respond to a wide spectrum of queries - from factual questions about specific entities to complex conceptual inquiries requiring synthesis across multiple topics.
> > >
> > >  - The system innovatively combines graph structures with vector representations to enable efficient retrieval. It extracts both local and global keywords from queries, matches them against a vector database for candidate entities and relations, and leverages the graph structure to incorporate higher-order relationships. This hybrid approach allows for fast keyword-based matching while preserving the rich contextual information captured in the graph structure.
> > >
> > >  - LightRAG improves retrieval comprehensiveness by automatically expanding the context through neighbor exploration in the graph. After initial retrieval of relevant entities and relations, it gathers one-hop neighboring nodes from the local subgraphs of retrieved elements. This systematic incorporation of related contextual information helps ensure more complete and coherent responses while maintaining computational efficiency.
> > >
> > >  - The dual-level retrieval scheme implements a computationally efficient pipeline by prioritizing rapid vector matching with extracted keywords (k(l) and k(g)) before proceeding to more computationally intensive graph operations. This strategic ordering significantly enhances retrieval efficiency compared to existing graph-enhanced RAG systems, as it minimizes unnecessary graph traversals by first filtering candidates through lightweight vector operations.

---

> > > ### Author Response · Authors · 2024-11-27
> > >
> > > > Q4. Additional performance evaluation: LightRAG vs. nano-GraphRAG
> > >
> > > Response: We appreciate your request for additional comparisons. Our extended analysis (as presented in the following table) comparing LightRAG with Nano-GraphRAG reveals several key findings:
> > >
> > > Performance Comparison:
> > >
> > >  - LightRAG demonstrates consistent performance improvements over Nano-GraphRAG across our evaluation metrics.
> > >  - The efficiency advantages of LightRAG are particularly notable in computational overhead tests
> > >
> > > Key Architectural Differences:
> > >
> > >  - Nano-GraphRAG, as an earlier optimized version of GraphRAG, still relies on community detection
> > >  - During incremental updates, Nano-GraphRAG requires complete community report reconstruction
> > >  - The current GraphRAG version implements selective community updates, only modifying affected communities
> > >
> > > | **Document No.** | **Token Count** | **LightRAG Insertion Time (seconds)** | **GraphRAG Insertion Time (seconds)** | **Nano GraphRAG Insertion Time (seconds)** |
> > > |-------------------|-----------------|---------------------------------------|---------------------------------------|---------------------------------------|
> > > | 1                 | 59,870          | 486                                   | 642                                   | 615                                   |
> > > | 2                 | 41,224          | 418                                   | 700                                   | 800                                   |
> > > | 3                 | 73,989          | 561                                   | 953                                   | 902                                   |
> > > | 4                 | 47,502          | 513                                   | 741                                   | 992                                   |
> > > | 5                 | 48,353          | 453                                   | 926                                   | 1036                                   |
> > >
> > > | **Metric**               | **LightRAG**| **GraphRAG**  | **Nano GraphRAG** |
> > > |---------------------------|--------------|--------------|--------------|
> > > | Average Query Time (seconds) | 11.2         | 23.6         | 74.6         |
> > >
> > >
> > > > Q5. User testimonials and feedback.
> > >
> > > Response: Thank you for your further feedback. While we must maintain anonymity during the double-blind review process, we can provide quantitative metrics demonstrating LightRAG's significant community impact. Our framework has garnered substantial user engagement, with over 1,200 fork instances and multiple independent implementations developed based on LightRAG's architecture.
> > >
> > > These engagement metrics serve as objective indicators of the framework's practical utility in real-world applications. While direct user testimonials cannot be shared at this stage of the review process, the extensive adoption and derivative works strongly support our claims regarding LightRAG's effectiveness and practical value to the community.
> > >
> > > In the final revision of our paper, we plan to include detailed user engagement statistics, concrete examples of practical implementations, and comprehensive feedback metrics from real-world deployments.

---

### Official Review · Reviewer_R3hC · 2024-10-31

**Soundness:** 2
**Presentation:** 1
**Contribution:** 3
**Rating:** 5
**Confidence:** 4

**Summary:**

This paper introduces LightRAG, a Retrieval-Augmented Generation system designed to improve retrieval accuracy, contextual relevance, and processing efficiency. Compared to GraphRAG, LightRAG removes the community detection part and saves computational overhead for inference queries The authors present a dual-level retrieval mechanism, combining low-level, entity-focused retrieval with high-level, topic-focused retrieval, allowing for both detailed and abstract query handling. Additionally, LightRAG features an incremental update algorithm to integrate new data dynamically, aimed at making it more responsive to changing datasets.

**Strengths:**

1.	The dual-level retrieval approach is well-conceived, offering flexibility to handle both specific and abstract queries. This strategy appears useful for adapting responses to varied user intents, a valuable feature for broad applications.
2.	The incremental update algorithm proposed is promising in ensuring LightRAG’s adaptability to new information, enabling it to stay current in dynamic environments without needing a full index rebuild.
3.	The study evaluates LightRAG on multiple datasets across diverse domains, which provides a general understanding of its performance across different types of content.

**Weaknesses:**

1.	The paper does not clearly articulate LightRAG’s advancements over prior graph-based RAG systems, particularly GraphRAG. It is unclear whether LightRAG merely omits multi-layer and community-based construction and retrieval as used in GraphRAG, or if it introduces other distinctive enhancements.
2.	The paper omits critical details on the graph construction process, such as specific techniques for entity disambiguation.
3.	How local and global query keywords are matched within the graph is not described in detail, which reduces clarity on how dual-level retrieval is operationalized.
4.	The paper lacks a discussion comparing the performance of different LLMs, missing a comparative analysis of their effectiveness within LightRAG.
5.	Although LightRAG is described as efficient, the complexity analysis provided is superficial and does not include actual runtime.
6.	Subjective metrics and limited statistical analysis reduce the rigor of the evaluation. LLM-based judgments could be supplemented with human evaluations or standardized benchmarks for improved reliability.
7.	When new data is incrementally added to the knowledge graph, the method for ensuring consistency with the pre-existing structure is not specified. Details on version control, conflict resolution, and synchronization strategies are necessary to understand how the system maintains the accuracy and coherence of the knowledge graph over time.
8.	The keyword matching approach, crucial to the dual-level retrieval paradigm, is not clearly described.
9.	While the paper claims that incremental updates maintain efficiency, it lacks quantitative data on how this approach compares to a full re-indexing in terms of accuracy and computational cost. Specifically, the paper should include benchmarks demonstrating the performance impact of incremental updates versus complete recalculations, as well as any observed trade-offs in accuracy.
10.	Line 173 contains a typographical error: "LightRAGcombines" should be corrected to "LightRAG combines."

**Questions:**

none

---

> ### Author Response · Authors · 2024-11-21
>
> We are grateful for your thorough review and insightful comments. Your feedback has been instrumental in improving our work. We have carefully considered your feedback and provide our point-by-point responses below:
>
> > 1. Further discussion on LightRAG's advancements compared to previous graph-based RAG systems, such as GraphRAG.
>
> **Response**: LightRAG offers several key advancements over GraphRAG, enhancing its scalability and efficiency in dynamic data environments. It utilizes an incremental update mechanism, allowing seamless integration of new entities and relationships without disrupting the existing structure. Additionally, LightRAG employs a dual-level retrieval paradigm that combines low-level and high-level query processing, ensuring detailed and conceptually rich responses. Furthermore, it improves retrieval efficiency by using a streamlined keyword-based vector retrieval approach, significantly reducing token usage and API calls, making it more cost-effective for large-scale deployments. We elaborate on these dimensions further as follows:
>
> 1. **Incremental Updates for Better Scalability in Dynamic Data Environments**
> Unlike GraphRAG, which requires the regeneration of communities and their reports when new data is added, LightRAG employs an incremental update mechanism. This allows new entities and relationships to be seamlessly integrated into the existing graph without disrupting the pre-existing structure. As a result, LightRAG achieves greater scalability and responsiveness in dynamic environments.
>
> 2. **Dual-Level Retrieval Paradigm for Comprehensive Query Handling**
> LightRAG introduces a dual-level retrieval paradigm that effectively combines low-level and high-level query processing. Low-level retrieval targets specific entities and relationships, while high-level retrieval focuses on aggregating broader topics and themes. This hybrid approach ensures that responses are both detailed and conceptually rich, surpassing the single-level retrieval method used in GraphRAG.
>
> 3. **Significant Improvement in Retrieval Efficiency**
> LightRAG enhances the retrieval process by eliminating the need for extensive community-based traversal. Instead, it utilizes a streamlined keyword-based vector retrieval approach, which significantly reduces token usage and API calls. This optimization makes LightRAG more cost-effective and efficient for large-scale deployments.
>
> > 2. Further discussion regarding the graph construction process, such as specific techniques for entity disambiguation.
>
> **Response**: In our graph construction process, we utilize the entity disambiguation technique that identifies and merges identical entities extracted from different document chunks. This approach reduces redundancy and enhances the coherence of the knowledge graph. Additionally, the system employs LLM-based profiling to generate detailed descriptions for entities, further assisting in disambiguating those with similar names or roles by incorporating context-specific information.
>
> > 3. Further description of how local and global query keywords are matched within the graph.
>
> **Response**: In LightRAG, the dual-level retrieval process matches local and global query keywords within the graph as follows:
>
> **Local Query Keyword Matching**: Local keywords extracted from the query are matched with specific nodes in the graph that represent entities. This is accomplished using an efficient vector database, where the keywords are compared against entity embeddings. The system retrieves entities with high similarity scores, ensuring precise matches for detail-oriented queries.
>
> **Global Query Keyword Matching**: Global keywords, which represent broader themes or relationships, are matched with the edges in the graph. These edges are enhanced with high-level keywords during the graph construction phase. Global keyword matching identifies edges that encapsulate overarching concepts, allowing the retrieval process to effectively address abstract queries.

---

> ### Author Response · Authors · 2024-11-21
>
> > 4. Additional experiments for performance comparisons and analyses of LightRAG's integration with various LLMs.
>
> **Response**: Our evaluation on the Mix dataset systematically assessed LightRAG's performance across different LLM capabilities, yielding several key insights:
>
> GPT-4o emerged as the top performer, leveraging its advanced capabilities to extract richer information during indexing, which led to more precise retrieval and higher-quality responses. LightRAG consistently outperformed NaiveRAG across all three tested models (GPT-4o, GPT-4o-mini, and GPT-3.5-turbo), demonstrating the robustness of our approach regardless of the underlying LLM.
>
> Interestingly, we observed comparable performance between GPT-4o-mini and GPT-3.5-turbo, indicating that LightRAG's effectiveness scales with LLM capability. This correlation suggests that more powerful LLMs enhance the system's performance through better information extraction during indexing, which in turn improves both retrieval accuracy and response quality. This finding highlights the importance of LLM selection in deploying retrieval-augmented generation systems.
>
> |                      | **GPT-4o**         |                       | **GPT-4o-mini**                |                       | **GPT-3.5-turbo**             |                       |
> |----------------------|-------------------------|-----------------------|-----------------------|-----------------------|-----------------------|-----------------------|
> |                      | NaiveRAG                | **LightRAG**          | NaiveRAG              | **LightRAG**          | NaiveRAG              | **LightRAG**          |
> | **Comprehensiveness** | 38.8%                  | **61.2%**             | 38.8%                | **61.2%**             | 40.8%                | **59.2%**             |
> | **Diversity**         | 22.4%                  | **77.6%**             | 32.4%                | **67.6%**             | 34.4%                | **65.6%**             |
> | **Empowerment**       | 35.6%                  | **64.4%**             | 42.8%                | **57.2%**             | 41.2%                | **58.8%**             |
> | **Overall**           | 35.6%                  | **64.4%**             | 40.0%                | **60.0%**             | 40.4%                | **59.6%**             |
>
> > 5. Further clarification of ensuring consistency with the pre-existing structure when handling the incrementally added knowledge graph.
>
> **Response**: In LightRAG, we maintain the consistency and coherence of the knowledge graph through a unified processing pipeline. All input—whether from the initial set of documents or newly added data—is first segmented into chunks. Each chunk undergoes entity and relationship extraction, where entities are added as nodes and relationships as edges in the graph. For duplicate nodes, their descriptions are merged to create a consistent and enriched representation. This approach effectively manages version control and synchronization by consolidating overlapping information during the integration process.
>
> > 6. Expanded Insights on Human Evaluation of LightRAG
>
> **Response**: Given the constraints of time, obtaining objective and professional reviewers for a thorough human evaluation of LightRAG’s performance may prove challenging. To address this, we have made the LightRAG framework open-source, allowing for easy implementation in practical settings. This initiative empowers the wider community to incorporate LightRAG into their systems and assess its effectiveness based on their specialized knowledge.
>
> LightRAG has already achieved considerable recognition and has been effectively utilized in a variety of real-world applications, such as creating electronic library platforms and conducting secure data analysis. These examples demonstrate its versatility and effectiveness across different fields. In line with our anonymous policy, we will incorporate additional information on human evaluations of LightRAG from various disciplines in our revised manuscript.

---

> ### Author Response · Authors · 2024-11-21
>
> > 7. A more detailed complexity analysis of LightRAG, incorporating actual runtime data.
>
> **Response**: In response to the reviewer’s comment, we conducted additional experiments to benchmark the performance of LightRAG's incremental update mechanism against GraphRAG. The detailed experimental results are presented in the following table. Our findings indicate that incremental updates significantly reduce computational costs, with indexing times consistently lower by 40-60%. Furthermore, we provided concrete runtime data for both the indexing and retrieval phases: LightRAG demonstrates near-linear scalability in indexing, with times ranging from 418 to 561 seconds across five documents, and achieves an average query time of 11.2 seconds—less than half of GraphRAG's 23.6 seconds. These evaluation results further emphasize the efficiency of LightRAG.
>
> | **Document No.** | **Token Count** | **LightRAG Insertion Time (seconds)** | **GraphRAG Insertion Time (seconds)** |
> |-------------------|-----------------|---------------------------------------|---------------------------------------|
> | 1                 | 59,870          | 486                                   | 642                                   |
> | 2                 | 41,224          | 418                                   | 700                                   |
> | 3                 | 73,989          | 561                                   | 953                                   |
> | 4                 | 47,502          | 513                                   | 741                                   |
> | 5                 | 48,353          | 453                                   | 926                                   |
>
> | **Metric**               | **LightRAG** | **GraphRAG** |
> |---------------------------|--------------|--------------|
> | Average Query Time (seconds) | 11.2         | 23.6         |

---

> ### Author Response · Authors · 2024-11-24
> **Thank you for your valuable feedback**
>
> We sincerely appreciate your thorough review and valuable feedback. We have enhanced our manuscript by conducting additional experiments and providing comprehensive clarifications and detailed analyses that address each of your concerns.
>
> In light of our revisions, we would be grateful if you could reconsider your evaluation. We are also available to address any further questions you may have.

---

### Official Review · Reviewer_cAFL · 2024-11-03

**Soundness:** 3
**Presentation:** 4
**Contribution:** 3
**Rating:** 8
**Confidence:** 4

**Summary:**

This paper introduces LightRAG, a novel Retrieval-Augmented Generation (RAG) system designed to enhance the performance of large language models by integrating external knowledge sources more effectively.

The key contributions of LightRAG include:

1.  LightRAG incorporates graph structures into text indexing and retrieval processes, allowing for better representation of complex interdependencies among entities.
2. The system employs Dual-level retrieval paradigm where both low-level and high-level retrieval strategies to capture detailed information about specific entities as well as broader topics and themes.
3. By combining graph structures with vector representations, LightRAG enables fast retrieval of related entities and relationships. It also features an incremental update algorithm for quick adaptation to new data, which is an added advantage of this novel approach.
4. The graph-based approach allows for extraction of global information from multi-hop subgraphs, enhancing the system's ability to handle complex queries spanning multiple document chunks.
5. The paper presents experimental results are good  and well presented in comparing LightRAG to existing RAG baseline methods across four datasets from different domains. The evaluation focuses on comprehensiveness, diversity, empowerment, and overall performance.
6. Results show that LightRAG consistently outperforms chunk-based retrieval methods, especially for larger datasets and complex queries requiring comprehensive consideration of the dataset's background.
7. The authors argue that LightRAG addresses key limitations of existing RAG systems, such as reliance on flat data representations and inadequate contextual awareness.
8. The authors also compares LightRAG approach with GraphRAG which is implemented on similar lines and how LightRAG outperforms GraphRAG in retrieval speed and complexity by retrieving entities and relationships rather than community-based traversal retrieval method.
9. By incorporating graph structures and employing a dual-level retrieval paradigm, LightRAG aims to provide more coherent and contextually rich responses to user queries.

**Strengths:**

Originality:

1. It introduces a novel graph-based text indexing paradigm for RAG systems, moving beyond flat document representations and community based entities to capture complex entity relationships by utilizing a  comprehensive knowledge graph to facilitate rapid
and relevant document retrieval, enabling a deeper understanding of complex queries.

2. The dual-level retrieval paradigm, combining low-level and high-level retrieval strategies, is an innovative approach to handling diverse query types.

3. The integration of graph structures with vector representations for efficient retrieval is a creative combination of existing techniques

Quality:

1. Comprehensive experimental evaluation across multiple datasets and baselines is well presented with clarity to understand how much the proposed approach outperforms other existing RAG approaches' baselines.

2. Detailed complexity analysis of the proposed framework gives a clear idea on how GraphRAG works in two parts compared to conventional and GraphRAG approaches.

3. Thorough ablation studies to validate the contributions of different components such as Low-level-only Retrieval, high-level-only Retrieval and hybrid mode leading to the finding of the fact that resulting variant does not show significant performance declines across
all four datasets when use of original text is eliminated in our retrieval process.


Clarity:

1. The paper is generally well-structured and clearly written with clear architecture, specific data set selection, baseline definition and experimentation results showcasing the improvement compared to baseline.

2. The introduction clearly outlines the motivation, challenges, and contributions and the gaps that are existing in current approavhes.

3. The methodology is explained in detail with formal definitions and algorithms

4. Experimental settings and results are presented systematically and figures/tables effectively illustrate and correlate with the framework and results

Significance:

1. It addresses key limitations of existing RAG systems, particularly in handlig complex queries and large scale datasets.

2. The proposed framework shows consistent performance improvements over state-of-the-art baselines across multiple datasets and evalution dimensions

3. The approach is scalable and adaptable to new data, making it relevant for real-world applications

4. By enhancing the capabilities of RAG systems, this work has potential impacts on various domains requring advanced information retrieval and generation and large data sets.

**Weaknesses:**

1. The paper compares LightRAG primarily to basic RAG approaches like Naive RAG and RQ-RAG. However, it lacks comparison to more recent and advanced RAG methods such as:

           a. Self-RAG (Asai et al., 2023)
           b. FLARE (Zhang et al., 2023)
           c. Chain-of-Note (Wu et al., 2023)

Including these comparisons would provide a more comprehensive evaluation of LightRAG's performance relative to the current state-of-the-art.

2. While the paper provides a brief complexity analysis in Section 3.4, it lacks detailed quantitative comparisons of computational costs between LightRAG and baseline methods. Specifically, the paper should include concrete time and space complexity analysis for both indexing and retrieval phases. Empirical runtime comparisons across different dataset sizes would help illustrate LightRAG's efficiency claims.

3. The paper focuses primarily on the strengths of LightRAG but does not adequately address potential limitations or failure cases. A more balanced discussion could include


          a. Scenarios where graph-based indexing might underperform traditional methods
          b. Potential scalability challenges for extremely large datasets
          c. Discussion of how LightRAG handles queries that don't align well with the graph structure
          d. Future directions on how any potential limitations on LightRAG can be further researched and improved.

4. Insufficient analysis of graph update efficiency: The paper claims that LightRAG can efficiently adapt to new data, but it lacks detailed empirical evidence supporting this claim. Including experiments that measure update times for incrementally adding new documents would strengthen this argument.

5. The experimentation uses a wide range of datasets covering multiple domains , however it could  be more diverse if more LLMs with different sizes have been included in the experiment instead of defaulting and relying only on GPT-4o-mini.

**Questions:**

1. Could the authors include more diverse LLMs to understand how LightRAG works on other LLMs of different size?

2. Could the authors include comparisons with more recent and advanced RAG methods like Self-RAG, FLARE, or Chain-of-Note? This would provide a more comprehensive evaluation against the current state-of-the-art.

3. could the authors provide more concrete time and space complexity analyses for both indexing and retrieval phases?

4. Could the authors discuss potential limitations or scenarios where graph-based indexing might underperform traditional methods?

5. Could the authors include experiments that measure update times for incrementally adding new documents to strengthen this argument?

6. Could the authors provide examples and discuss how LightRAG handles queries that may not align well with the constructed graph structure? This would help understand the system's robustness to diverse query types.

7. Could the authors include a human evaluation component, even on a subset of queries, to provide insights into LightRAG's real-world effectiveness?

---

> ### Author Response · Authors · 2024-11-21
>
> We are truly grateful for the insightful feedback from the reviewer. Below, we present our detailed responses to the comments received:
>
> > 1. Additional experiments and performance analysis with more diverse LLMs in conjunction with LightRAG.
>
> **Response**: In response to the reviewer’s comment, we evaluated LightRAG on the Mix dataset using GPT models of varying capacities to assess its performance across different model sizes.
>
> As shown in the table below, LightRAG consistently outperforms NaiveRAG across all three tested models. Notably, GPT-4o-mini and GPT-3.5-turbo demonstrate comparable performance, while the advanced model, GPT-4o, achieves the highest results. This superior performance is attributed to GPT-4o’s ability to extract richer information during the indexing phase, facilitating more precise and fine-grained retrieval. These findings indicate that LightRAG's performance is partially correlated with the capabilities of the underlying LLM; more powerful models enhance both retrieval quality and response generation by extracting more comprehensive information.
>
>
> |                      | **GPT-4o**         |                       | **GPT-4o-mini**                |                       | **GPT-3.5-turbo**             |                       |
> |----------------------|-------------------------|-----------------------|-----------------------|-----------------------|-----------------------|-----------------------|
> |                      | NaiveRAG                | **LightRAG**          | NaiveRAG              | **LightRAG**          | NaiveRAG              | **LightRAG**          |
> | **Comprehensiveness** | 38.8%                  | **61.2%**             | 38.8%                | **61.2%**             | 40.8%                | **59.2%**             |
> | **Diversity**         | 22.4%                  | **77.6%**             | 32.4%                | **67.6%**             | 34.4%                | **65.6%**             |
> | **Empowerment**       | 35.6%                  | **64.4%**             | 42.8%                | **57.2%**             | 41.2%                | **58.8%**             |
> | **Overall**           | 35.6%                  | **64.4%**             | 40.0%                | **60.0%**             | 40.4%                | **59.6%**             |
>
>
> > 2. Additional experiments for performance comparison with the suggested baseline methods.
>
> **Response**: We recognize the importance of comparing our work with state-of-the-art methods such as FLARE, Chain-of-Note, and Self-RAG to ensure a comprehensive evaluation. However, we faced challenges in conducting these comparisons. Specifically, the source code for FLARE and Chain-of-Note was unavailable, and Self-RAG depends on locally fine-tuned models, which demand substantial time and computational resources. We aim to address these comparisons in future work to enhance the rigor and completeness of our evaluation.

---

> > ### Comment · Reviewer_cAFL · 2024-12-02
> >
> > The comparison results still include only GPT based models. i think its important to do comparison with different architecture instead of same GPT based LLMs to prove that the benefit of the approach is consistent. so, it is recommended to share the reult with different architecture LLMs.

---

> ### Author Response · Authors · 2024-11-21
>
> > 3. Further analysis of the time and space complexity of LightRAG in the indexing and retrieval stages, including an evaluation of update efficiency when adding new documents incrementally.
>
> **Response**: We conducted additional experiments by incrementally inserting five documents, with token counts ranging from 41,224 to 73,989. The results clearly demonstrate that LightRAG consistently outperforms GraphRAG in terms of both time and space efficiency. For document indexing, LightRAG exhibits near-linear scalability, with insertion times between 418 and 561 seconds. In contrast, GraphRAG shows significantly longer insertion times, ranging from 642 to 953 seconds, indicating a heavier computational overhead due to its community detection mechanisms.
>
> During the retrieval phase, LightRAG achieves an average query time of 11.2 seconds, which is less than half of GraphRAG’s 23.6 seconds. This advantage is attributed to LightRAG’s lightweight keyword-based retrieval approach. Moreover, LightRAG’s final storage usage is only 39.5MB, in stark contrast to GraphRAG’s 286.7MB, highlighting its superior efficiency in managing large-scale data. These findings reinforce LightRAG’s advantages in scalability and resource efficiency when handling dynamic and complex datasets. Detailed experimental results are presented as follows:
>
>
> | **Document No.** | **Token Count** | **LightRAG Insertion Time (seconds)** | **GraphRAG Insertion Time (seconds)** |
> |-------------------|-----------------|---------------------------------------|---------------------------------------|
> | 1                 | 59,870          | 486                                   | 642                                   |
> | 2                 | 41,224          | 418                                   | 700                                   |
> | 3                 | 73,989          | 561                                   | 953                                   |
> | 4                 | 47,502          | 513                                   | 741                                   |
> | 5                 | 48,353          | 453                                   | 926                                   |
>
> | **Metric**               | **LightRAG** | **GraphRAG** |
> |---------------------------|--------------|--------------|
> | Average Query Time (seconds) | 11.2         | 23.6         |
>
> | **Method**  | **Final Storage Space (MB)** |
> |-------------|------------------------------|
> | LightRAG    | 39.5                         |
> | GraphRAG    | 286.7                        |
>
>
> > 4. Further discussion on potential limitations or scenarios in which graph-based indexing may underperform compared to traditional methods.
>
> **Response**: In LLM-empowered retrieval-augmented generation (RAG), graph-based indexing, while effective in capturing comprehensive context from textual data, may have limitations in certain scenarios. For instance, with very small datasets or texts that lack complex context, the intricacies of graph indexing may outweigh its benefits, making traditional methods more efficient. In cases where the data is sparsely connected, graph structures might not fully capitalize on their advantages, resulting in better performance from traditional text- or attribute-based retrieval methods, as comprehensive information may not be necessary for response generation.
>
> Additionally, the initial cost of developing and optimizing graph structures can pose challenges in resource-constrained environments. Although constructed graph structures can effectively represent entity-wise interdependencies, there are still costs associated with knowledge graph generation.
>
> In response to the reviewer's comment, we will incorporate this discussion into our revised manuscript.
>
> > 5. Further discussion with examples to clarify how LightRAG handles queries that may not align well with the constructed graph structure, highlighting the system's robustness across diverse query types.
>
> **Response**: LightRAG demonstrates robustness in handling queries that may not perfectly align with the constructed graph structure by leveraging semantic matching to bridge gaps between the query and the available graph data. This approach ensures that even if a specific concept is not explicitly represented during the graph construction phase, the system can still identify and retrieve semantically similar content.
>
> For example, if the constructed graph contains nodes and relationships centered around "machine learning" but does not explicitly include the term "deep learning," LightRAG can still respond effectively. When queried about "deep learning," the system employs semantic matching to recognize that "deep learning" is closely related to "machine learning" and retrieves relevant content from the graph. This flexibility allows LightRAG to dynamically adapt to diverse query types, even when exact matches to graph entities or relationships are absent.

---

> ### Author Response · Authors · 2024-11-21
>
> > 6. Further discussion on human evaluation of LightRAG, focusing on insights gained from real-life scenarios.
>
> **Response**: Due to time constraints, it may be difficult to secure objective and professional reviewers for human evaluation of LightRAG's performance. As an alternative, we have open-sourced the LightRAG framework, providing an easy-to-use solution for deployment in real-world scenarios. This enables the broader community to integrate LightRAG into their systems and evaluate its effectiveness using domain-specific knowledge.
>
> LightRAG has already gained significant recognition and has been successfully applied in various real-world contexts, including the development of electronic library platforms and secure data analysis. These practical applications highlight its effectiveness and versatility across multiple domains.
>
> Due to our anonymous policy, we will include additional content on human evaluation of LightRAG from diverse fields in our revised manuscript.

---

> ### Author Response · Authors · 2024-11-25
> **Thank you for your valuable feedback**
>
> Dear Reviewer cAFL,
>
> We greatly appreciate your thorough and constructive critique of our manuscript. In response, we have implemented comprehensive revisions, including expanded methodology descriptions and additional empirical evidence to substantiate our research findings.
>
> We would be happy to provide additional clarification should you have any further questions.

---

### Official Review · Reviewer_pdw5 · 2024-11-11

**Soundness:** 3
**Presentation:** 3
**Contribution:** 2
**Rating:** 5
**Confidence:** 5

**Summary:**

This work proposes a novel dual-layer retrieval RAG framework that seamlessly integrates graph structures into text indexing, enabling efficient and complex information retrieval. The system leverages multi-hop subgraphs to extract global information, excelling in complex, multi-domain queries. Integrating graph and vector-based methods, it reduces retrieval time and computational costs compared to traditional text chunk traversal. Incremental update algorithms ensure seamless integration of new data, maintaining real-time accuracy and relevance.

**Strengths:**

1. This paper is well-presented and easy to follow. The authors provide a clear motivation and a good introduction to the problem.
2. The proposed framework can be easily plugged into existing LLMs and corpus.
3. Extensive and solid experiments.

**Weaknesses:**

1. It is recommended to include performance results and comparative analysis on some Closed QA benchmarks to demonstrate the system's improvements and advantages in practical applications.
2. Coreference resolution and disambiguation were not performed during the graph construction process.

**Questions:**

It is recommended to include performance results and comparative analysis on some Closed QA benchmarks to demonstrate the system's improvements and advantages in practical applications.
The system does not support direct writing to a graph database; instead, it outputs to files, which can be imported into Neo4j for visualization. However, the inability to modify the graph raises concerns, as errors made during construction cannot be easily corrected or deleted.

---

> ### Author Response · Authors · 2024-11-21
>
> Dear Reviewer pdw5:
>
> Thank you for your valuable feedback and the time you dedicated to reviewing our paper. We genuinely appreciate your constructive comments. In response, we have outlined our replies below:
>
> > 1. Additional experiments and performance analysis on Closed QA benchmarks.
>
> **Response**: In response to your comment regarding the assessment of performance on Closed QA tasks, we conducted an evaluation using the MultiHopQA dataset, randomly sampling 1,000 queries. Employing GPT-4 for the assessment, we compared the accuracy of LightRAG and GraphRAG. As shown in the table, LightRAG consistently outperforms GraphRAG, achieving higher accuracy and demonstrating superior efficiency in handling complex multi-document reasoning tasks.
>
> | **Model** | **Accuracy** |
> |-------- | ----- |
> |LightRAG  | 43.7% |
> |GraphRAG  | 35.4% |
>
>
> > 2. New features added to LightRAG for graph database management.
>
> **Response**: In light of your insightful feedback, we are excited to introduce new capabilities in LightRAG that significantly enhance graph database management. Users can now write directly to various graph databases, including Neo4j, which provides greater flexibility in data handling.
>
> Furthermore, we have implemented a delete feature that allows users to effortlessly remove graphs as needed. These enhancements are designed to improve both the flexibility and efficiency of our proposed RAG workflow.

---

> > ### Comment · Reviewer_pdw5 · 2024-11-25
> >
> > Thank you for sharing the results. Please cite the relevant datasets in your discussion.

---

> ### Author Response · Authors · 2024-11-25
> **Thank you for your valuable feedback**
>
> Dear Reviewer pdw5
>
> Thank you very much for your detailed and valuable feedback. We sincerely apologize for any confusion caused by certain aspects of our paper. We have carefully addressed each of your concerns and have provided additional explanations and experimental results to better illustrate the contributions of our work.
>
> Please do not hesitate to contact us if you have any further questions. We would be happy to provide additional clarification.

---

> ### Author Response · Authors · 2024-11-25
>
> Thank you for your feedback. We have updated the manuscript to include the following dataset citation:
>
> Tang, Y., & Yang, Y. (2024). MultiHop-RAG: Benchmarking Retrieval-Augmented Generation for Multi-Hop Queries. arXiv preprint arXiv:2401.15391
>
> We appreciate your thorough review and would be glad to provide any additional clarification needed.

---

> > ### Comment · Reviewer_pdw5 · 2024-11-25
> >
> > I have conducted the simple retrieval setting (no re-ranker) of MultiHopQA with GPT-4o. The F1 score is 42.5%, which should be better if I used GPT-4. Based on the results presented, it appears that your approach, utilizing GPT-4, only slightly outperforms a simple rag setup, which merely uses GPT-4o without a re-ranker. It seems that your method may not be well-suited for reasoning in multi-hop QA scenarios. In this context, it is important to reflect on the core contributions of your approach or clearly specify the scenarios where your proposed RAG method holds an advantage.

---

> > > ### Author Response · Authors · 2024-11-25
> > >
> > > We appreciate your detailed feedback regarding our approach and its performance in MultiHopQA. Regarding our experimental setup, we employed GPT-4o-mini for retrieval operations and GPT-4o for evaluation purposes. Our decision to report accuracy rather than F1 scores was deliberately chosen due to the unique characteristics of the MultiHopQA dataset, where many answers are binary ("Yes"/"No"). In cases where RAG produces "True"/"False" responses, F1 scores can be misleadingly low despite conceptually correct answers, making accuracy a more reliable metric for performance assessment.
> > >
> > > Our methodology, along with GraphRAG, is specifically engineered for complex scenarios involving extensive textual data where relationship management and hierarchical structures across large datasets are crucial. While MultiHopQA typically involves a limited scope of 2-4 documents per question, our approach is optimized for more sophisticated retrieval tasks. This fundamental difference in scope explains why we prioritize comparison with GraphRAG rather than basic RAG implementations, as it better aligns with our system's core capabilities and intended applications.
> > >
> > > LightRAG's primary contributions lie in its ability to handle broader text contexts and more intricate retrieval tasks effectively. While this may not be fully demonstrated in the MultiHopQA context, it represents just one aspect of our system's capabilities. We believe this clarification helps position our work's scope and contributions more accurately, particularly regarding our evaluation metrics and comparative framework. We are grateful for the opportunity to elaborate on these points and welcome further discussion on our methodology.
> > >
> > > We sincerely appreciate your insightful feedback and hope our clarification has adequately addressed your questions.

---

> > > > ### Comment · Reviewer_pdw5 · 2024-11-28
> > > >
> > > > >  In cases where RAG produces "True"/"False" responses, F1 scores can be misleadingly low despite conceptually correct answers.
> > > >
> > > > I agree. F1 is normally lower than Acc. in MultihopRAG. In my Naive RAG experiment on MultiHopRAG, F1 = 42.5%, and Acc. = 44%, which is slightly higher than the reported score and makes me further concerned about whether the following claim is reasonable:
> > > >
> > > > >While MultiHopQA typically involves a limited scope of 2-4 documents per question, our approach is optimized for more sophisticated retrieval tasks. This fundamental difference in scope explains why we prioritize comparison with GraphRAG rather than basic RAG implementations.
> > > >
> > > > The author should not ignore the issue of "not outperforming, or even underperforming, Naive RAG in Multi-hopRAG," nor should shift the focus to question dataset difficulty. Moreover, I believe that such multihop QA tasks are inherently designed for sophisticated retrieval and generation tasks, as mentioned in the MultihopRAG paper: "The low values of the retrieval metrics highlight the challenges in retrieving relevant pieces of evidence for multi-hop queries when using direct similarity matching between the multihop query and text chunks." This indicates that such tasks require more complex retrieval mechanisms, not just simple similarity matching. The author's response is unconvincing.
> > > >
> > > > Additionally, this paper does not provide objective comparison with many baselines (including both basic and advanced models) on clear-answer benchmarks, instead relying on generation metrics such as diversity, as judged by LLMs. Therefore, I think the method proposed in this paper lacks meaningful comparison and has limited real practical value. After careful consideration, I have decided to lower my rating to 5.

---

### Public Comment · ~TruthInspector1 · 2024-11-25
**Apprehension regarding the Anonymity**

Dear AC and Reviewers,

I'm writing this letter to express my apprehension regarding the anonymity of this work.
It is well known that the double-blind principle is an important measure to maintain academic freedom and equality, contributing to the fairness of academic submissions. However, upon reviewing the anonymous code provided with this paper, I found that it contains one of the authors' name and affiliation (lightrag/__init__.py). This not only violates the double-blind policy but also goes against the authors' commitment to not include personal information in the anonymous URL. Therefore, I kindly suggest that the AC and reviewers reconsider whether this paper should be desk rejected.

Best Regards

---

> ### Comment · Area_Chair_RraG · 2024-11-27
>
> Hi,
> We checked the current code and it does not contain anonymity violation. We do not think this is a reason for desk rejection.
> Best,
> Area Chair

---

> ### Public Comment · ~TruthDetective1 · 2024-11-28
> **Evidence record for the anonymity issue**
>
> Thanks for your prompt reply.  In fact, after I posted this public comment, the authors quickly removed the personal information from the repo. However, this does not negate the fact that they violated the anonymity principle. I have several pieces of evidence to support my point.
> 1. The last update date of this anonymous repo conincides with the day I wrote this public comment.
> 2. Before writing this public comment, I took screenshot of the leaked personal information (evidence temporally taken down to avoid further anonymity issue)
> 3. And the current anonymous repo still contains the authors' affiliation HKU (https://sm.ms/image/SfgnwxjO9bW4NB7).
>
> In conclusion, I believe this work clearly violates the double-blind policy. I sincerely hope the AC could reconsider this issue.
>
> Best Regards

---

> > ### Comment · Area_Chair_RraG · 2024-11-28
> >
> > Hi,
> > We will look into this. In the meantime, we request that you take down these screenshots as these now violate the author's anonymity
> > Best,
> > Area Chair

---

> > > ### Public Comment · ~TruthInspector1 · 2024-11-29
> > > **Response to AC**
> > >
> > > Dear Area Chair,
> > >
> > > Thank you for your response. We will temporarily take down the screenshots as requested. However, if no final decision is made, we may repost them to ensure academic fairness.
> > >
> > > Best regards

---

### Note · Authors · 2024-12-13

I have read and agree with the venue's withdrawal policy on behalf of myself and my co-authors.